# HybridVLA: Collaborative Diffusion and Autoregression in a Unified Vision-Language-Action Model

**Jiaming Liu**[1], **Hao Chen**[2*], **Zhuoyang Liu**[1*], **Pengju An**[1], **Renrui Zhang**[2†], **Chenyang Gu**[1], **Xiaoqi Li**[1], **Ziyu Guo**[2], **Sixiang Chen**[1,3], **Mengzhen Liu**[1,3], **Chengkai Hou**[1], **Mengdi Zhao**[4], **Kaichen Zhou**[1], **Pheng-Ann Heng**[2], **Shanghang Zhang**[1] ✉

[1]State Key Laboratory of Multimedia Information Processing, School of Computer Science, Peking University; [2] The Chinese University of Hong Kong ; [3] BAAI;   [4] Fudan University

* Equal contribution, † Project lead, ✉ Corresponding author; Project web page

## Abstract

A central objective of manipulation policy design is to enable robots to comprehend human instructions and predict generalized actions in unstructured environments. Recent autoregressive vision-language-action (VLA) approaches discretize actions into bins to exploit the pretrained reasoning and generation paradigms of vision-language models (VLMs). While these models achieve efficient and scalable training, the discretization undermines the continuity required for precise control. In contrast, diffusion-based VLA methods incorporate an additional diffusion head to predict continuous actions, but they rely solely on feature representations extracted from the VLM, without leveraging the pretrained large language model (LLM) as an expert for iterative action generation. To integrate the complementary strengths of autoregressive and diffusion generation, we introduce HybridVLA, which innovatively leverages a shared LLM backbone to perform iterative action prediction through both paradigms. Specifically, a collaborative training recipe is proposed, incorporating diffusion denoising into the next-token prediction process and mitigating interference between the two generation paradigms. With this recipe, we find these two action prediction methods not only reinforce each other but also exhibit varying strengths across different scenarios. Therefore, we design a collaborative action ensemble mechanism that adaptively fuses both predictions, leading to more robust control. HybridVLA outperforms previous state-of-the-art VLA methods by 17% and 19% in mean success rate on simulation and real-world tasks, respectively, while demonstrating generalization to unseen configurations.

## 1 Introduction

Developing intelligent robots capable of performing manipulation tasks demands robust policies (Driess et al., 2023; Huang et al., 2023). In dynamic and unstructured real-world environments, such policies need to interpret human instructions and generalize across a wide range of complex tasks. Recently, vision-language models (VLMs) (Alayrac et al., 2022; Li et al., 2023a) have achieved significant breakthroughs in common-sense reasoning, primarily driven by advances in model architecture, large-scale pretraining, and the iterative generation paradigm. Building on this success, several studies have extended VLMs into vision-language-action (VLA) models, enabling them to predict low-level action poses for robotic manipulation (Brohan et al., 2023; Kim et al., 2024). This paradigm outlines a promising roadmap for building foundation models to facilitate generalist robots.

On the one hand, autoregressive VLA methods (Li et al., 2024b; Kim et al., 2024) emulate the pretrained reasoning and generation paradigms of VLMs for next action-token prediction, enabling efficient and scalable training (Pertsch et al., 2025). These methods enable generalized action prediction by quantizing continuous actions into discrete bins that occupy part of the LLM's original vocabulary. However, this discretization disrupts the continuity of action poses and hinders precise

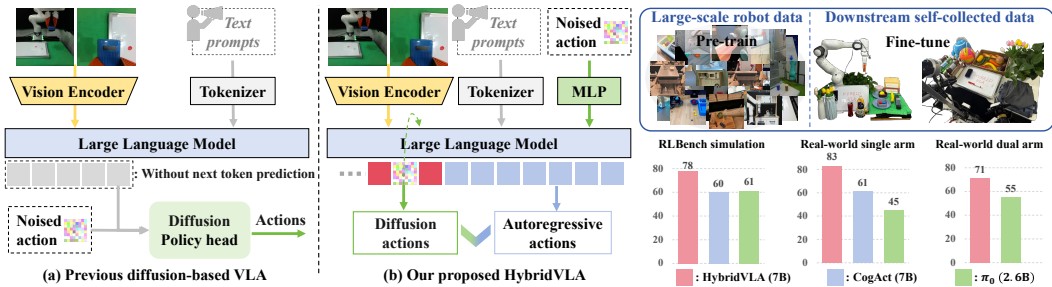

Figure 1: **(a)** Unlike recent diffusion-based VLA methods that attach a separate diffusion head after VLMs, **(b)** HybridVLA innovatively integrates diffusion and autoregressive action prediction within a single LLM, embedding the denoising process of diffusion into the next-token prediction. Under our proposed methods, HybridVLA achieves remarkable performance across a wide range of tasks.

control (Wen et al., 2024a). On the other hand, building on the success of diffusion models in content generation (Ho et al., 2022; Peebles & Xie, 2023), diffusion policies have been introduced in robotic imitation learning (Chi et al., 2023b; Reuss et al., 2023; Xian et al., 2023). Recent diffusion-based VLA methods (Black et al., 2024; Li et al., 2024a; Wen et al., 2024a; Bjorck et al., 2025) incorporate a diffusion head after the VLM, leveraging probabilistic denoising for action prediction. While these methods enable precise manipulation, the diffusion head lacks internet-scale pretraining and depends solely on feature representations extracted from the VLM, without fully leveraging the powerful LLM backbone as an action expert for iterative generation. Given these advantages and limitations, a question arises: "*How can we elegantly construct a unified VLA model that integrates the strengths of both autoregressive and diffusion policies?*"

To this end, we propose HybridVLA, which leverages a unified LLM backbone to perform both autoregressive and diffusion action generation, harnessing the complementary strengths of both paradigms for robust robot control. Unlike prior diffusion-based VLA methods (Black et al., 2024; Li et al., 2024a) that append an independent diffusion head after the LLM (Figure 1 (a)), we introduce a collaborative training recipe that embeds the Markovian denoising steps of diffusion into the next-token prediction process (Figure 1 (b)), enabling each step to be interpreted as a reasoning iteration within the pretrained LLM. To stabilize the joint optimization of the autoregressive and diffusion components, we design a robotics-specific token sequence formulation that organizes multimodal inputs, diffusion tokens, and autoregressive tokens through specialized markers. Under this recipe, HybridVLA captures continuous action representations from diffusion modeling while inheriting the pretrained reasoning paradigm of autoregression, enabling the two paradigms to jointly approximate the same conditional action distribution. Empirically, these action prediction methods not only reinforce each other but also exhibit varying strengths across different tasks. Therefore, a collaborative action ensemble mechanism is proposed, where the two predictions are adaptively fused based on autoregressive action token confidence, improving robustness in manipulation.

To enhance generalization capability, we initialize HybridVLA with a pretrained VLM (Karamcheti et al., 2024) and adopt a step-by-step training strategy. As shown in the right of Figure 1, HybridVLA is first pretrained on large-scale, diverse, cross-embodiment robotic datasets, including Open X-Embodiment (O'Neill et al., 2023), DROID (Khazatsky et al., 2024), and ROBOMIND (Wu et al., 2024b), covering 760K trajectories and over 10K A800 GPU training hours. It is then fine-tuned on self-collected simulation data (James et al., 2020) and real-world demonstrations, achieving state-of-the-art (SOTA) manipulation performance across a wide range of tasks with both single-arm and dual-arm robots. In real-world testing, HybridVLA also exhibits strong generalization to unseen objects, backgrounds, spatial layouts, and lighting conditions, underscoring the effectiveness of our collaborative model design and training recipe. Moreover, we demonstrate that the autoregressive discrete action outputs of HybridVLA can be replaced with language-based task planning without compromising the stability of diffusion-based action prediction. Our contributions are as follows:

- We propose HybridVLA, which innovatively leverages a single LLM backbone for iterative action prediction through both autoregressive and diffusion generation within a unified token sequence, harnessing the complementary strengths of both paradigms.
- We introduce a collaborative training recipe that embeds the denoising process of diffusion into next-token prediction, enabling mutual reinforcement of both generation paradigms.

Additionally, we propose a collaborative action ensemble mechanism that adaptively fuses autoregressive and diffusion-based actions, enhancing manipulation robustness.

- Our proposed HybridVLA achieves SOTA performance across diverse tasks while demonstrating strong generalization to several unseen configurations.

## 2 RELATED WORK

**Vision-language-action (VLA) models.** Some studies (Ahn et al., 2022; Driess et al., 2023; Huang et al., 2023; 2024b) enable robots to interpret both language and visual observations, automatically generating task plans. Meanwhile, vision-language-action (VLA) models leverage the inherent reasoning abilities of VLMs to predict low-level SE(3) poses. Specifically, RT2 (Brohan et al., 2023) quantizes 7-DoF actions into discrete bins for autoregressive pose prediction. Building on this, ManipLLM (Li et al., 2024b) incorporates affordance priors through chain-of-thought reasoning, while OpenVLA (Kim et al., 2024) performs large-scale pretraining on the Open X-Embodiment dataset (O'Neill et al., 2023). FAST (Pertsch et al., 2025) applies the discrete cosine transform to enable fast and scalable training of autoregressive-based VLA models. To support continuous action prediction, some VLA approaches (Liu et al., 2024a; Huang et al., 2024a; Li et al., 2023b; Wu et al., 2023a) incorporate a policy head, such as an MLP or LSTM (Graves & Graves, 2012), and use regression loss for imitation learning. However, quantization in autoregressive methods disrupts action continuity, while regressive methods fail to incorporate probabilistic action representations.

**Diffusion-based VLA models.** Building on the success of diffusion models in content generation (Ho et al., 2020; 2022; Peebles & Xie, 2023), diffusion policies have been applied in robotics (Chi et al., 2023a), including reinforcement learning (Ajay et al., 2022; Wang et al., 2022), imitation learning (Pearce et al., 2023; Prasad et al., 2024; Reuss et al., 2023; Xian et al., 2023), grasping (Simeonov et al., 2023; Urain et al., 2023; Wu et al., 2023b), and motion planning (Janner et al., 2022; Saha et al., 2024). Following this, 3D Diffusion Actor (Ke et al., 2024) and DP3 (Chi et al., 2023b) employ diffusion models to interpret point cloud data. Octo (Team et al., 2024) and RDT-1B (Liu et al., 2024b) augment a transformer for diffusion modeling to predict flexible actions. To integrate diffusion with VLMs, $\pi_0$ (Black et al., 2024) and $\pi_{0.5}$ (Intelligence et al., 2025) add an expert head that generates actions through flow matching, while TinyVLA (Wen et al., 2024b) incorporates a simple diffusion head after the lightweight VLM. CogACT (Li et al., 2024a) and DiVLA (Wen et al., 2024a) decouple reasoning and action prediction into the VLM and an injected diffusion head, respectively. Following this architecture, some works (Bjorck et al., 2025; Bu et al., 2025; figureai) introduce a dual-system design to enable control at different frequencies. However, in these methods, the diffusion head operates as a separate module and treats the VLM as a multimodal feature extractor, limiting its ability to fully exploit the pretrained knowledge of VLM. Unlike prior methods focused on image and language generation quality (Ge et al., 2024; Wu et al., 2024a;c; Xie et al., 2024), HybridVLA introduces a robotics-specific collaborative training strategy. Meanwhile, CDP (Ma et al., 2025) provides long-horizon conditioning for future action prediction, and ARP (Zhang et al., 2025) predicts task-specific action chunks to balance action accuracy with generation efficiency. In contrast, HybridVLA focuses on unifying diffusion-based and autoregressive action generation within the unified model and token sequence, enabling the two paradigms to mutually enhance each other.

## 3 HYBRIDVLA METHOD

**Problem Statement.** At time $t$, each demonstration consists of image observations $o_t$, language description $l_t$, and the current robot state $r_t$. Our model $\pi$ aims to predict action $a$ to control the robot arms, which can be formulated as: $\pi : (o_t, l_t, r_t) \rightarrow a_{t+1:t+H}$, H is the action horizon. Following Kim et al. (2024), the action $a$ represents the end-effector pose, which uses 7-DOF and 14-DOF for single-arm and dual-arm control, respectively. Each 7-DOF action includes 3-DOF for relative translation offsets ($[\Delta x, \Delta y, \Delta z] \in \mathbb{R}^3$), 3-DOF for rotation (Euler angles $\in \mathbb{R}^3$), and 1-DOF for the gripper state (open/closed $\in \mathbb{R}^1$). The ground truth (GT) and the model-predicted action are in SE(3), formulated as: $a = [\Delta x, \Delta y, \Delta z, Roll, Pitch, Yaw, 0/1]$.

**Motivation.** First, existing diffusion-based VLA methods (Black et al., 2024; Li et al., 2024a) append a separate diffusion head after the VLM and further train it to predict continuous actions. However,

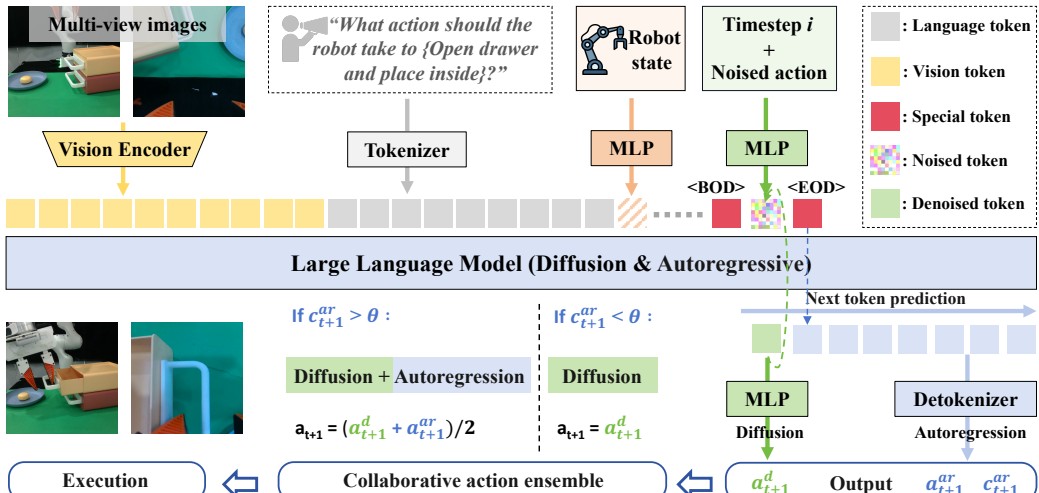

Figure 2: **HybridVLA Framework.** All multimodal inputs are encoded into tokens and subsequently organized into our designed token sequence formulation within the LLM's embedding space. For diffusion tokens, HybridVLA simultaneously projects the denoising timestep and noise into continuous vector representations. The corresponding noisy samples are iteratively fed into the LLM to predict the noise at each step. The marker tokens, <BOD> (Beginning of Diffusion) and <EOD> (End of Diffusion), are introduced to bridge the two generation paradigms. Subsequently, autoregressive actions are generated via next action-token prediction, explicitly conditioned on the preceding tokens. The two actions can be adaptively ensembled for robot arm control.

such diffusion heads lack internet-scale pretraining and rely solely on features extracted from the VLM as conditions, without leveraging the LLM backbone as an action expert for iterative generation. Second, the autoregressive and diffusion paradigms offer distinct strengths in VLA modeling. Diffusion-based predictions excel at precise manipulation, particularly in fine-grained control and tasks involving dynamic objects. Autoregressive predictions, by inheriting the VLM generation paradigm, learn more efficiently from demonstrations (Pertsch et al., 2025; Intelligence et al., 2025) and show superior ability to understand flexible instructions and unseen objects. Empirical evidence supporting these insights is presented in Section A.1. Therefore, we propose HybridVLA, which leverages a unified LLM backbone to perform iterative action prediction through both autoregressive and diffusion generation, integrating the complementary strengths of both paradigms.

## 3.1 HYBRIDVLA ARCHITECTURE

This section presents the architecture and workflow of HybridVLA, which is available in two model sizes based on 7B and 2.7B large language models (LLMs). Following Kim et al. (2024), both Hybrid-VLA (7B) and HybridVLA (2.7B) inherit the base architecture from Prismatic VLMs (Karamcheti et al., 2024), initializing with the corresponding pretrained VLM parameters. We then present the two basic components, the vision encoders and the LLM backbone, as shown in Figure 2.

**Vision encoders.** HybridVLA leverages powerful vision encoder combinations, such as DI-NOv2 (Oquab et al., 2023) and SigLIP (Zhai et al., 2023), to capture semantic features $f_d \in \mathbb{R}^{B \times N_v \times 1024}$ and $f_s \in \mathbb{R}^{B \times N_v \times 1152}$. $B$ and $N$ represent batch size and token sequence length, respectively. These features are concatenated along the channel dimension to form $f_v \in \mathbb{R}^{B \times N_v \times 2176}$, which is subsequently projected into the LLM's embedding space via a projection layer. Hybrid-VLA(2.7B) uses only the CLIP (Radford et al., 2021) model as its vision encoder. When processing multi-view images, we use the shared vision encoders to extract features from each view, which are then concatenated along the token dimension.

**LLM backbone.** HybridVLA adopts the 7B LLaMA-2 (Touvron et al., 2023a) as its LLM, which is responsible for multimodal understanding and action generation. Language prompts are encoded into the embedding space $f_l \in \mathbb{R}^{B \times N_l \times 4096}$ using the pretrained tokenizer, then concatenated with visual tokens and fed into the LLM. For HybridVLA (2.7B), the workflow remains identical to that of HybridVLA (7B) but employs the 2.7B Phi-2 (Javaheripi et al., 2023) as the LLM.

Table 1: Exploration of token sequence formulations. All models are trained using hybrid objectives. Dif and AR refer to using only autoregressive or diffusion-based generation on 10 RLBench tasks.

| | Type 1 | Type2 | Type3 | **Type4 (ours)** |
|---|---|---|---|---|
| Dif | 0.67 | 0.56 | 0.65 | **0.72** |

| | Type 1 | Type2 | Type3 | **Type4 (ours)** |
|---|---|---|---|---|
| AR | 0.59 | 0.54 | 0.60 | **0.65** |

## 3.2 COLLABORATIVE TRAINING RECIPE

To better integrate both diffusion and autoregressive generation capabilities within the LLM's next-token prediction process, we propose a collaborative training strategy that includes a unified token sequence formulation, hybrid objectives, and structured training stages.

**Token sequence formulation design.** As shown in Figure 2, this design aims to organize multimodal tokens within the LLM's embedding space into a unified and ordered token sequence, enabling coordination between the two generation paradigms during the next-token prediction process. In addition to the acquired vision and language tokens, our framework also integrates the robot state, diffusion timestep, noisy actions, and the autoregressive tokens. For the **robot state**, we integrate it into the LLM to enhance temporal consistency in action generation. Instead of discretizing the robot state and merging it with the language query (Li et al., 2024b) (Type 1 of Table 1), we employ a learnable MLP to map the robot state directly into the embedding space, $f_r \in \mathbb{R}^{B \times 1 \times 4096}$. For diffusion-based actions, we predict them through a diffusion denoising process. During training, the **diffusion timestep** and **noisy actions** are projected into the LLM's embedding space through MLPs, represented as continuous vectors. To seamlessly connect diffusion-related tokens within this token sequence, we introduce special beginning-of-diffusion (<BOD>) and end-of-diffusion (<EOD>) tokens to encapsulate them. This design not only clarifies the boundaries between diffusion and autoregressive generation but also prevents confusion in the next-token prediction process, such as avoiding diffusion tokens directly predicting masked discrete tokens (Type 2 of Table 1). For **autoregressive actions**, we quantize the end-effector pose into discrete bins and replace part of the vocabulary in the LLM (Kim et al., 2024), which is then tokenized into a sequence of discrete tokens. Due to the autoregressive nature of LLMs (Touvron et al., 2023b), both the question and the answer, including the discrete action ground truth (GT), are provided during training, whereas only the question is available at inference time. Therefore, placing autoregression before the diffusion tokens may cause action GT leakage (Type 3 in Table 1), as all preceding tokens serve as conditions in diffusion modeling. To avoid this, we position diffusion tokens before autoregression to explicitly provide continuous latent conditions for subsequent token prediction (Type 4 in Table 1). Moreover, since diffusion operates on noise, it naturally circumvents the risk of information leakage.

**Hybrid objectives.** To equip HybridVLA with both autoregressive and diffusion generative capabilities, we combine two training losses under our designed token sequence. For the *diffusion* part, we adopt the standard objective used in diffusion policies (Chi et al., 2023b), which minimizes the mean squared error between the predicted noise $\epsilon_\pi$ and the sampled Gaussian noise $\epsilon$. The corresponding loss function is defined as: $\mathcal{L}_{dif} = E_{a,i,c}||\epsilon - \epsilon_\pi(a_t^i, i, c)||^2$, where $\epsilon \sim \mathcal{N}(0, 1)$ and $c$ represents the conditioning context. For the *autoregressive* part, we minimize the cross-entropy loss $\mathcal{L}_{ar}$ to train the model on predicting the discrete actions. Under our proposed token sequence formulation, both loss functions are jointly optimized in a unified training objective, defined as: $\mathcal{L}_{hybrid} = \mathcal{L}_{dif} + \mathcal{L}_{ar}$. For action generation, the autoregressive and diffusion branches aim to approximate the same action distribution space, as the action data are normalized in the same way (range [-1, 1]). Note that the discrete action is simply a quantized representation of this distribution. Moreover, we validate that the hybrid objectives, coupled with our proposed token formulation, foster mutual reinforcement between the two generation paradigms, as evidenced by both quantitative experiments (Section 4.2) and Principal Component Analysis (Appendix A.2).

**Structured training stage.** After loading the pretrained VLM parameters, HybridVLA undergoes two training stages with hybrid objectives: large-scale pretraining on open-source robotic data and fine-tuning on self-collected data. During pretraining, we train HybridVLA for 10 epochs on 35 datasets. These datasets contain 760K robot trajectories, comprising 33M frames. Due to dataset differences, pretraining relies solely on single 2D observations, whereas fine-tuning relies on either single or multi-view observations, depending on the downstream task. The details of the pretraining dataset are provided in Appendix B.1.

### 3.3 COLLABORATIVE ACTION ENSEMBLE

During inference, either autoregressive or diffusion-based actions can be used for robot control. However, we observe that the two prediction methods not only reinforce each other but also exhibit varying strengths across different tasks. This motivates a collaborative action ensemble mechanism that adaptively fuses both predictions.

**Diffusion actions.** When generating diffusion actions, we append the special token <BOD> after the preceding condition tokens to signal the start of the denoising process. We employ DDIM (Song et al., 2020) with $n$ sampling steps, and find empirically that $n$ can be reduced to as low as 4 while maintaining an optimal balance between performance and inference speed. As shown in the right part of Figure 2, at each denoising step, only the current noisy sample is input to the LLM to predict the noise for the next step, and the token sequence does not retain any previous noise samples. Each denoising step is treated as a reasoning iteration, allowing HybridVLA to progressively refine diffusion-based action predictions by leveraging the LLM's pretrained knowledge. In this way, we enable a multi-step Markovian denoising process that aligns with the LLM's next-token prediction mechanism. After obtaining the denoised tokens, we use an MLP to map them to the action space. To accelerate sampling, we introduce a KV cache design for the diffusion process. During the initial step, the model processes the vision and language tokens, the denoising timestep, and the initial noise. In subsequent steps, only the updated timestep and noisy actions are forwarded, while the cached keys and values are reused. This approach reduces redundant computation and significantly enhances inference efficiency for diffusion-based action generation.

**Autoregressive actions.** As shown in Figure 2, the autoregressive generation begins after the special token <EOD>. Unlike previous autoregressive VLA methods (Kim et al., 2024), HybridVLA's autoregressive generation additionally conditions on continuous action representations derived from diffusion tokens. This yields superior manipulation performance over standalone autoregressive paradigms without explicit continuous latent conditioning, as shown in the ablation study.

**Ensembled actions.** After obtaining the two types of actions under our collaborative training recipe, we empirically observe two phenomena. 1) Different action types demonstrate varying performance across tasks and scenarios. 2) The confidence of autoregressive tokens serves as a reliable indicator of action quality. In over 80% of successfully completed test samples, the average confidence of autoregressive action tokens exceeds 0.96 (range [0, 1]). Therefore, as shown in Figure 2, we use the mean confidence of autoregressive tokens ($c^{ar}_{t+1}$) to guide the action ensemble. If the confidence exceeds $\theta$ ($\theta = 0.96$), we consider the autoregressive action ($a^{ar}_{t+1}$) sufficiently accurate and perform an average operation with the diffusion action ($a^d_{t+1}$). Otherwise, we rely solely on the diffusion action to control the robot. Further analysis of the confidence threshold can be found in Appendix C.2.

## 4 EXPERIMENT

In Section 4.1, we compare the manipulation ability of HybridVLA with previous VLA methods in simulation environments. The effectiveness of each component is validated in Section 4.2 and Appendix C.2. In Section 4.3, we present both quantitative and qualitative results of HybridVLA in real-world scenarios. The generalization capabilities of HybridVLA are examined in Section 4.4, testing on unseen manipulated instances, background, spatial positions, and lighting conditions.

### 4.1 SIMULATION EXPERIMENT

**Simulation benchmark.** To systematically evaluate our method, we select the RLBench (James et al., 2020) benchmark in the CoppeliaSim simulator, which contains 10 different tabletop tasks. These

Table 2: **Comparison of HybridVLA and baselines on RLBench.** We train all methods in the multi-task setting (Shridhar et al., 2022) and report the success rates (S.R.) and variances (Var.).

| Models | Close box | Close laptop lid | Toilet seat down | Sweep to dustpan | Close fridge | Phone on base | Umbrella out | Frame off hanger | Wine at rack | Water plants | Mean S.R. ↑ & Var. ↓ | Infer. speed ↑ |
|---|---|---|---|---|---|---|---|---|---|---|---|---|
| ARP (one view) | 0.35 | 0.60 | 0.75 | 0.80 | 0.70 | 0.30 | 0.40 | 0.25 | 0.35 | 0.20 | 0.47 ±0.03 | - |
| ARP (four views) | 0.55 | **0.95** | **1.00** | **0.90** | 0.95 | 0.45 | 0.50 | 0.40 | 0.70 | 0.40 | 0.68 ±0.02 | - |
| ManipLLM (7B) | 0.50 | 0.80 | 0.40 | 0.20 | 0.80 | 0.35 | 0.10 | 0.25 | 0.15 | 0.20 | 0.38 ±0.05 | 2.2 Hz |
| OpenVLA (7B) | 0.65 | 0.40 | 0.75 | 0.60 | 0.80 | 0.20 | 0.35 | 0.15 | 0.10 | 0.10 | 0.41 ±0.02 | 6.3 Hz |
| OpenVLA-OFT (7B) | **1.00** | 0.65 | 0.60 | 0.30 | 0.80 | 0.30 | 0.30 | 0.20 | 0.20 | 0.15 | 0.45 ±0.03 | 13.4 Hz |
| $\pi_0$ (2.6B) | 0.90 | 0.60 | **1.00** | 0.30 | 0.90 | 0.25 | 0.35 | **0.75** | 0.65 | 0.45 | 0.61 ±0.03 | 13.8 Hz |
| CogACT (7B) | 0.80 | 0.85 | 0.90 | 0.65 | 0.90 | 0.50 | **0.60** | 0.35 | 0.25 | 0.25 | 0.60 ±0.04 | 9.8 Hz |
| **HybridVLA-ar (7B)** | 0.90 | 0.90 | 0.95 | 0.85 | 0.95 | 0.30 | 0.30 | 0.40 | 0.45 | 0.50 | 0.65 ±0.04 | 6.3 Hz |
| **HybridVLA-dif (7B)** | 0.95 | 0.90 | **1.00** | 0.55 | 0.90 | 0.25 | 0.55 | **0.75** | **0.85** | 0.45 | 0.72 ±0.03 | 9.4 Hz |
| **HybridVLA (7B)** | 0.95 | **0.95** | **1.00** | **0.90** | **1.00** | **0.55** | **0.60** | 0.70 | 0.60 | **0.55** | **0.78** ±0.04 | 6.1 Hz |
| **HybridVLA (2.7B)** | **1.00** | 0.90 | 0.90 | 0.80 | 0.90 | 0.25 | 0.55 | 0.45 | 0.70 | 0.25 | 0.67 ±0.03 | 12.3 Hz |

tasks are performed using a Franka Panda robot and a front-view camera. The data are collected using pre-defined waypoints and the Open Motion Planning Library (Sucan et al., 2012). Following the frame-sampling method used in previous works (Shridhar et al., 2022; Goyal et al., 2023; Jia et al., 2024), we construct the training dataset, with each task consisting of 100 trajectories. We further validate our approach on the SimplerEnv (Li et al., 2024c), with details provided in Appendix C.1.

**Training and Evaluation Details.** We compare our method with five previous SOTA VLA models, including ManipLLM (Li et al., 2024b), OpenVLA (Kim et al., 2024), OpenVLA-OFT (Kim et al., 2025), $\pi_0$ (Black et al., 2024), CogACT (Li et al., 2024a). Meanwhile, we also compare our method with the related work ARP (Zhang et al., 2025). Specifically, we report two versions: ARP (four views), which uses four camera views, and ARP (one view), which uses only the front-view camera, matching the camera configuration used by other VLA methods. To ensure a fair comparison, we load the official pretrained parameters provided by each method, adhering to their respective training settings. We categorize our method into four modes: HybridVLA-ar (7B), HybridVLA-dif (7B), HybridVLA (7B), and HybridVLA (2.7B). All modes are jointly trained using our proposed collaborative training recipe. However, HybridVLA-ar (7B) and HybridVLA-dif (7B) rely solely on autoregressive or diffusion-based action generation during inference, respectively. For HybridVLA, the single-view RGB input is resized to $224 \times 224$, and the robot state is consistent with predicted actions (7-DOF end-effector poses). Our models are trained for 300 epochs on downstream tasks using mixed-precision. For evaluation, following previous VLA method (Kim et al., 2024), we test all methods with 20 rollouts per task from the latest epoch checkpoint, repeating the process three times to report the mean success rate with variance.

**Quantitative Results.** As shown in Table 2, HybridVLA (7B) achieves an average success rate of 78% across 10 distinct tasks, outperforming the previous SOTA autoregressive-based VLA (OpenVLA) and diffusion-based VLA ($\pi_0$) by 37% and 17%, respectively. These results demonstrate that our method effectively integrates the two generation approaches within a shared LLM backbone, simultaneously capturing the continuous characteristics of diffusion-based actions and inheriting the LLM's pretrained generation paradigm for efficient learning from demonstrations. Remarkably, compared to CogACT and $\pi_0$, HybridVLA-dif (7B) also achieves performance improvements of 12% and 11%, respectively. These results highlight that, unlike previous approaches which attach the diffusion head after the VLM, our method more effectively leverages the LLM's pretrained knowledge to fully unlock the potential of diffusion action prediction. Note that by manually annotating sub-task plans and applying GPT (Achiam et al., 2023) for automated augmentation, we train HybridVLA (7B) to generate language-based plans autoregressively and actions through diffusion. This training paradigm achieves a task success rate of 74%, not only validating the effectiveness of our proposed collaborative generation method but also demonstrating that the autoregressive generation branch of HybridVLA does not compromise the stability of diffusion-based action prediction. Finally, HybridVLA (2.7B) delivers satisfactory results, confirming our method's effectiveness in enhancing VLM manipulation capabilities across different backbone sizes. **For inference speed**, As shown in Table 2, when tested on an NVIDIA 4090D GPU, HybridVLA-dif (7B) and HybridVLA (2.7B) achieve satisfactory model inference speed comparable to CogACT (7B) and $\pi_0$ (2.6B). Note that all models are run with bfloat16 precision during inference, without employing action chunking.

## 4.2 ABLATION STUDY

**The impact of each component.** We conduct ablation experiments on 10 RLBench tasks, using the same training and evaluation settings as in the simulation experiments. **For collaborative training**

**recipe**, we begin by comparing different **token sequence formulation** designs in Table 1, demonstrating that Type 4 yields the best performance. The corresponding analysis is provided in Section 3.2.

To validate the effectiveness of **hybrid objectives** under our proposed token formulation, we present a comparative study in Table 3, contrasting Ex1 with Ex2, and Ex3 with Ex4. Specifically, Ex1 and Ex3 are trained using only the autoregressive loss $\mathcal{L}_{ar}$ or diffusion loss $\mathcal{L}_{dif}$, respectively, and thus produce only the corresponding action type. In contrast, Ex2 (HybridVLA-ar) and Ex4 (HybridVLA-dif) are both trained with the hybrid loss $\mathcal{L}_{hybrid}$, yet are constrained to output only autoregressive or diffusion actions, respectively. These results validate that our proposed hybrid training not only avoids negative interference between the

Table 3: **Impact of each component.** AR and Dif denote that use solely autoregressive and diffusion-based action, respectively. CAE indicates the collaborative action ensemble method, whereas LSP refers to large-scale pretraining on robotic datasets.

| | AR | Dif | CAE | $\mathcal{L}_{ar}$ | $\mathcal{L}_{dif}$ | $\mathcal{L}_{hybrid}$ | LSP | Mean ↑ |
|---|---|---|---|---|---|---|---|---|
| Ex1 | ✓ | - | - | ✓ | - | - | ✓ | 0.57 |
| Ex2 | ✓ | - | - | - | - | ✓ | ✓ | 0.65 |
| Ex3 | - | ✓ | - | - | ✓ | - | ✓ | 0.65 |
| Ex4 | - | ✓ | - | - | - | ✓ | ✓ | 0.72 |
| Ex5 | ✓ | ✓ | ✓ | - | - | ✓ | ✓ | 0.78 |
| Ex6 | ✓ | ✓ | ✓ | - | - | ✓ | - | 0.22 |

two generation paradigms but also enables mutual reinforcement. Finally, the comparison between Ex5 and Ex6 highlights the importance of the **structured training stage**. Although Ex6 is initialized with pretrained VLM parameters, it suffers from a significant drop in accuracy, highlighting the essential role of large-scale pretraining on robot datasets in ensuring stable control. **For collaborative action ensemble**, as evidenced by the results of Ex2, Ex4, and Ex5 in Table 3, the performance of HybridVLA (Ex5) is further improved, which demonstrates that fusing the two output modes enhances the robustness of robot control. Moreover, the confidence of the autoregressively generated action can be used as an indicator to guide the fusion of actions from the two paradigms. The above ablation studies corroborate our initial motivation that the two action-generation paradigms possess distinct advantages, and HybridVLA effectively integrates them during both training and inference. Due to space limitations, Appendix C.2 provides additional ablation studies on: (1) confidence thresholds in the collaborative action ensemble, (2) the influence of the diffusion-based KV cache on inference speed, and (3) the impact of DDIM sampling steps on performance.

Table 4: **The impact of different confidence threshold.** We report success rates for HybridVLA (7B) and HybridVLA (2.7B) on various tasks with confidence threshold from 0.90 to 0.98.

| Threshold | Close box | Close laptop lid | Toilet seat down | Sweep to dustpan | Close fridge | Phone on base | Umbrella out | Frame off hanger | Wine at rack | Water plants | Mean S.R. ↑ |
|---|---|---|---|---|---|---|---|---|---|---|---|
| **HybridVLA (7B)** | | | | | | | | | | | |
| 0.90 | 0.80 | 0.85 | 0.95 | 0.95 | 0.85 | 0.50 | 0.40 | 0.55 | 0.55 | 0.45 | 0.68 |
| 0.92 | 0.95 | 0.85 | 1.00 | 0.90 | 0.90 | 0.40 | 0.40 | 0.70 | 0.60 | 0.45 | 0.72 |
| 0.94 | 0.95 | 0.90 | 1.00 | 0.90 | 0.95 | 0.55 | 0.50 | 0.65 | 0.55 | 0.50 | 0.75 |
| 0.96 | 0.95 | 0.95 | 1.00 | 0.90 | 1.00 | 0.55 | 0.60 | 0.70 | 0.60 | 0.55 | **0.78** |
| 0.98 | 0.95 | 0.90 | 0.95 | 0.90 | 0.95 | 0.55 | 0.50 | 0.70 | 0.55 | 0.45 | 0.74 |
| **HybridVLA (2.7B)** | | | | | | | | | | | |
| 0.90 | 0.70 | 0.75 | 0.85 | 0.80 | 0.90 | 0.25 | 0.45 | 0.40 | 0.50 | 0.10 | 0.58 |
| 0.92 | 0.85 | 0.90 | 0.90 | 0.80 | 0.85 | 0.25 | 0.45 | 0.35 | 0.50 | 0.20 | 0.61 |
| 0.94 | 1.00 | 0.85 | 0.95 | 0.75 | 0.85 | 0.25 | 0.40 | 0.40 | 0.60 | 0.25 | 0.63 |
| 0.96 | 1.00 | 0.90 | 0.90 | 0.80 | 0.90 | 0.25 | 0.55 | 0.45 | 0.70 | 0.25 | **0.67** |
| 0.98 | 0.90 | 0.90 | 0.95 | 0.55 | 0.90 | 0.20 | 0.55 | 0.35 | 0.70 | 0.15 | 0.62 |

**The impact of confidence threshold in collaborative action ensemble.** We evaluated HybridVLA on ten RLBench tasks, varying the confidence threshold from 0.90 to 0.98. The score of each task under every confidence threshold and different backbones are shown in Table 4. We find that when the confidence threshold drops below 0.94, autoregressive predictions become unreliable, leading to a slight degradation in the performance of the ensemble action. Conversely, when the threshold reaches 0.98, the number of valid autoregressive actions becomes too limited, causing the performance of the ensemble action to closely match that of the diffusion-predicted action. Empirically, we conclude that setting the threshold to 0.96 ensures a stable action ensemble. For different backbones, we conduct the same ablation experiment using the 2.7B Phi-2 model as the LLM backbone and find that setting the action-token confidence threshold to 0.96 still serves as a robust indicator for determining whether action ensembling should be applied.

Table 5: **Real-world experiments.** The manipulation success is determined by human evaluation. Since CogACT lacks support for multi-view images, which are crucial for dual-arm tasks (Black et al., 2024; Fu et al., 2024), we conduct our dual-arm comparison solely with $\pi_0$.

| Models | Franka single-arm robot | | | | | | AgileX dual-arm robot | | | | | |
|---|---|---|---|---|---|---|---|---|---|---|---|---|
| | Pick and place | Unplug charger | Pour water | Wipe blackboard | Open drawer and place inside | Mean. S.R. ↑ | Pick and place | Lift ball and place | Place bottles at rack | Wipe blackboard | Fold shorts | Mean. S.R. ↑ |
| $\pi_0$ (2.6B) | 0.50 | 0.35 | 0.45 | 0.35 | **0.60** | 0.45 | 0.75 | 0.65 | 0.40 | 0.30 | 0.65 | 0.55 |
| CogACT (7B) | 0.80 | 0.70 | 0.40 | 0.65 | 0.50 | 0.61 | - | - | - | - | - | - |
| **HybridVLA(7B)** | **0.90** | **0.95** | **0.80** | **0.85** | 0.65 | **0.83** | **0.90** | **0.80** | **0.60** | **0.55** | **0.70** | **0.71** |

Single-arm real-world tasks      Dual-arm real-world tasks

## 4.3 REAL-WORLD EXPERIMENT

**Self-collected Data.** For single-arm tasks, we use a Franka Research 3 robot with a static front-view and a wrist-view camera. We perform 5 tasks: 1) *Pick and place*, 2) *Unplug charger*, 3) *Open drawer and place object inside*, 4) *Pour water*, 5) *Wipe blackboard using eraser*. For each task, 100 demonstrations are collected using a SpaceMouse device. For dual-arm tasks, we use an AgileX robot equipped with a static exterior view, a right-wrist view, and a left-wrist view camera. We conduct 5 coordinated dual-arm tasks: 1) *Pick and place*, 2) *Lift ball and place it in basket*, 3) *Place two bottles at rack*, 4) *Wipe blackboard using eraser*, 5) *Fold shorts*. Similarly, 100 demonstrations are collected for each task using master-puppet teleoperation. Additional details are provided in Appendix B.2.

**Training and Evaluation Details.** We evaluate HybridVLA (7B) against previous VLA methods, $\pi_0$ (Black et al., 2024) and CogACT (Li et al., 2024a). The implementation details remain consistent with our simulation experiments, except for using two-view inputs for single-arm tasks and three-view inputs for dual-arm tasks. For evaluation, we use the checkpoint from the latest epoch to perform 20 rollouts across diverse tabletop positions.

**Quantitative and Qualitative Results.** In Table 5, HybridVLA achieves outstanding performance across single-arm tasks. For *Pick and place* and *Unplug charger*, HybridVLA achieves success rates of 90% and 95%, respectively, demonstrating accurate object position prediction. For *Pour water*, HybridVLA outperforms the previous SOTA method by 35%, showcasing its ability to comprehend object relationships and predict precise rotations. The superior performance on *Wipe blackboard* and *Open drawer and place inside* further underscores the robustness of our method in long-horizon tasks. For dual-arm tasks, we extend the action dimensions of both diffusion and autoregressive tokens to 14-DOF, representing the 7-DOF end-effector poses for both the right and left arms. Our method consistently outperforms previous VLA approaches across five distinct tasks, highlighting HybridVLA's ability to effectively leverage LLM's pretrained knowledge for dual-arm coordination in complex scenarios. Furthermore, in the lower part of Table 5, we present visualizations of the manipulation processes performed by our method, which accurately predict actions across various task demands, including precise positioning and rotation, dual-arm coordination, and scene understanding. Additional qualitative results and failure case analyses are provided in Appendix D and Appendix E, respectively, and execution videos are available in the supplementary materials.

## 4.4 GENERALIZATION EXPERIMENT

Since CogACT and $\pi_0$ excel in single-arm and dual-arm tasks, respectively, we design four common generalization experiments, comparing our HybridVLA (7B) with CogACT on the single-arm *Pick*

Table 6: **Generalization.** "Object", "Background", "Height", and "Lighting" denote unseen manipulated objects, backgrounds, spatial positions, and lighting conditions, respectively. The image on the left depicts the unseen test scenarios, with red boxes marking the key differences.

| Scenario | Pick and place (single arm) | | Lift ball and place (dual arm) | | Pick and Place (dual arm) | | Place Bottles on Rack (dual arm) | |
|---|---|---|---|---|---|---|---|---|
| | HybridVLA | CogACT | HybridVLA | $\pi_0$ | HybridVLA | $\pi_0$ | HybridVLA | $\pi_0$ |
| Original | 0.90 | 0.80 | 0.80 | 0.65 | 0.90 | 0.75 | 0.60 | 0.40 |
| Object | 0.60(-33%) | 0.45(-43%) | 0.75(-6%) | 0.60(-8%) | 0.90(-0%) | 0.55(-26%) | 0.55(-8%) | 0.30(-25%) |
| Background | 0.80(-11%) | 0.50(-37%) | 0.60(-25%) | 0.50(-23%) | 0.80(-11%) | 0.50(-33%) | 0.50(-17%) | 0.30(-25%) |
| Height | 0.75(-17%) | 0.50(-37%) | 0.60(-25%) | 0.45(-31%) | 0.70(-22%) | 0.50(-33%) | 0.45(-25%) | 0.25(-37%) |
| Lightning | 0.70(-22%) | 0.60(-25%) | 0.75(-6%) | 0.55(-15%) | 0.80(-11%) | 0.65(-13%) | 0.55(-8%) | 0.35(12%) |
| Mean | 0.71(-21%) | 0.51(-36%) | 0.68(-15%) | 0.52(-20%) | 0.80(-11%) | 0.55(-27%) | 0.51(-15%) | 0.30(-25%) |

*and place* task and with $\pi_0$ on the *dual-arm Pick and Place*, *Lift ball and place*, and *Place Bottles on Rack* task. The visualization of four generalization test scenarios is shown in the left part of Table 6.
**1) Unseen manipulated objects.** In this scenario, we replace the training manipulated objects with a series of unseen objects, e.g., replacing the red block with a charger. As shown in the "Object" row of Table 6, our method demonstrates the smallest accuracy drop. These results indicate that HybridVLA effectively integrates diffusion into the autoregressive next-token prediction process, not only capturing the continuous characteristics of diffusion-based generation, but also preserving the object-level semantic reasoning capabilities of the pretrained VLM. **2) Unseen background.** In this scenario, cluttered backgrounds are introduced during testing, such as adding unseen flowers around the manipulated object. HybridVLA still shows satisfactory results, further demonstrating that our proposed training recipe effectively inherits the VLM's scene-level understanding, enhancing robustness to environmental variations. **3) Unseen Spatial position.** Unlike position shifts within the same plane, we introduce height variations during testing, further challenging the model's spatial comprehension. As shown in the "Height" row of Table 6, HybridVLA consistently achieves precise manipulation even when encountering objects in previously unseen spatial positions. These results highlight that HybridVLA exhibits strong trajectory generalization through the ensemble of two action generation methods. **4) Unseen lighting conditions.** Finally, we introduce variations in lighting conditions, a common challenge in real-world environments. All methods maintain satisfactory performance, demonstrating that large-scale pretraining on robotic datasets enhances their generalization across diverse data distributions. To provide a clearer overview, in the Table 6 below, we summarize the average score and average accuracy drop percentage across all unseen configurations. The results show that our method reduces the accuracy drop by approximately 5–16% compared to the baselines under generalization scenarios. These findings demonstrate that HybridVLA effectively integrates diffusion into the autoregressive next-token prediction, achieving not only more robust action generation, but also more efficient learning from demonstrations, thereby enhancing its generalization capability across diverse tasks.

## 5 CONCLUSION AND LIMITATION

In this paper, we introduce HybridVLA, a unified Vision-Language-Action (VLA) framework that equips a single LLM with both diffusion-based and autoregressive action generation capabilities. To integrate the distinct strengths of both paradigms, we propose a collaborative training recipe that embeds diffusion denoising into the next-token prediction process, enabling mutual reinforcement and improving manipulation robustness. By effectively inheriting the continuous nature of diffusion-based action generation and leveraging the pretrained knowledge of LLMs, HybridVLA achieves outstanding performance and strong generalization across both simulation and real-world tasks. One limitation of HybridVLA is that its inference speed is constrained by the slower autoregressive generation, similar to prior autoregressive VLA methods (Kim et al., 2024; Brohan et al., 2023; Li et al., 2024b). However, our collaborative training enables mutual reinforcement between the two generation methods, allowing inference using only the diffusion-based action for robot control (HybridVLA-dif), achieving a 9.4 Hz inference speed.

ACKNOWLEDGEMENTS

This work was supported by the National Natural Science Foundation of China (62476011). This work was also supported by the National Natural Science Foundation of China (625B2007).

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

**Appendix A.** To validate our motivation, we first present empirical analyses showing that the autoregressive and diffusion action generation paradigms exhibit distinct advantages across different tasks and scenarios. Furthermore, through Principal Component Analysis, we demonstrate that our collaborative training recipe enables mutual reinforcement between the two paradigms.

**Appendix B.** We present the details of our large-scale pretraining and self-collected real-world datasets.

**Appendix C.** Additional simulation experiments and ablation studies are presented.

**Appendix D.** We include further visualizations of both single-arm and dual-arm manipulation processes.

**Appendix E.** An analysis of failure cases encountered when using HybridVLA to control a robot.

**Appendix F.** The Use of Large Language Models (LLMs)

## A  VALIDATION OF THE MOTIVATION

### A.1  DISTINCT STRENGTHS OF TWO GENERATION PARADIGMS

In this section, we present a variety of experimental comparisons to highlight the respective advantages of the autoregressive and diffusion action output paradigms, while also emphasizing the necessity of integrating the two generation approaches. To this end, we employ two modes of our model: Our-ar and Our-dif, which use only autoregressive or diffusion-based action generation during inference, respectively.

**Fine-grained task.** We evaluate Our-ar and Our-dif on a fine-grained manipulation task (unplug a charging cable from its docking base) using an AgileX dual-arm robot. Under the same training and testing setup as Section 4.3, both models were trained on 30 demonstrations. As shown in Figure 3(a), Our-dif achieves significantly higher action accuracy than Our-ar, which is critical for fine-grained control. We attribute this superiority to the continual action generation nature of diffusion and our proposed method, which allows the diffusion process to more effectively exploit the pretrained knowledge of the LLM through progressively refined action predictions.

**Dynamic manipulation task.**  To evaluate the two action generation paradigms on a dynamic manipulation task, we conducted a pick-and-place experiment with the AgileX robot under controlled perturbations. Specifically, the target banana was dynamically shifted left or right within the left arm's manipulable range prior to pick-up (see the first row of Figure 10). We compared Our-dif and Our-ar directly using their trained models in a zero-shot manner. As shown in Figure 3 (b), Our-dif achieves a higher success rate, underscoring its superior robustness in dynamic manipulation scenarios.

**Unseen objects.** Following the setup of the generalization experiments (Section 4.4), we evaluate Our-dif and Our-ar using single-arm robot on a pick-and-place task where the manipulated objects are replaced with previously unseen instances (e.g., a charger or a strawberry). As shown in the Figure 3 (c), Our-ar undergoes a smaller performance degradation upon object replacement, suggesting that the autoregressive paradigm is more effective at capturing semantic variations across novel objects.

**Unseen language instructions.** Since the RLBench benchmark provides multiple language instructions for each task, we directly conduct simulator experiments using unseen instructions to test Our-ar and Our-dif. For each task, we employ a variety of semantically equivalent instructions that were not encountered during training. As shown in Figure 3(d), the performance of Our-ar decreases by only 9% on average, which is much smaller than that of Our-dif. This demonstrates that the autoregressive paradigm exhibits relatively robust contextual reasoning ability when handling flexible natural language instructions.

Consequently, these results highlight a clear pattern: diffusion-based generation excels at producing fine-grained, temporally consistent actions, particularly in dynamically evolving environments, whereas autoregressive action generation inherits the large-scale pretrained paradigm of VLMs, enabling more efficient demonstration learning (Pertsch et al., 2025; Intelligence et al.) and exhibiting robustness in language comprehension and generalization to novel objects. Building upon this observation, we propose HybridVLA. Our approach leverages a unified LLM backbone to generate

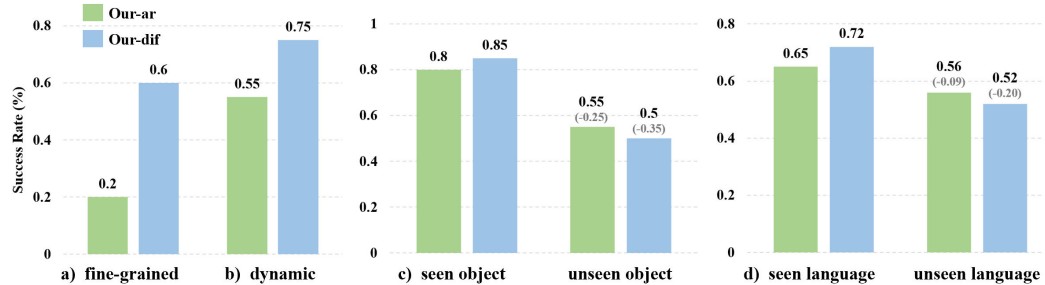

Figure 3: **Respective strengths of diffusion-based and autoregressive action generation paradigms.** We evaluate the performance of Our-ar and Our-dif across a variety of scenarios.

actions through both autoregressive and diffusion paradigms, thereby harnessing the distinct strengths of each.

## A.2 PRINCIPAL COMPONENT ANALYSIS

In Section 4.2, we validate the effectiveness of our Collaborative Training Recipe through ablation studies in the simulator, demonstrating that joint training with hybrid objectives enables mutual reinforcement between the two generation paradigms, compared to training them individually. To validate that our Collaborative Training Recipe improves the representation capacity of both action generation paradigms, we follow Xiao et al. (2024) and conduct a Principal Component Analysis (PCA) study of their feature distributions. In particular, we sample several trajectories from both Pick and Place actions and feed the corresponding frames into the model. From these inputs, we extract the diffusion-denoised tokens as well as the autoregressive action tokens, and project them into a 2D space using PCA. We compare models trained with our collaborative training recipe against models where each generation paradigm is trained independently, i.e., optimized solely with either the diffusion loss or the autoregressive loss. As shown in Table 7, jointly trained models yield diffusion and autoregressive features that form tighter intra-class clusters and exhibit larger inter-class separation for both Pick and Place actions. This indicates that joint optimization not only improves the feature representation, but also implicitly regularizes the latent space to preserve dimensions beneficial to both diffusion- and autoregressive-based generation.

Table 7: **PCA feature analysis of HybridVLA.** Comparison of intra-class and inter-class distances under collaborative training versus independent training. Collaborative optimization yields tighter intra-class clustering and larger inter-class separation.

| Metric | Our Collaborative Training | | Independent Training | |
|---|---|---|---|---|
| | Diffusion Token | AR Token | Diffusion Token | AR Token |
| Intra-class Distance | 0.49 | 0.44 | 0.73 | 0.91 |
| Inter-class Distance | 8.7 | 10.8 | 8.6 | 4.4 |

## B ADDITIONAL DATASET DETAILS

### B.1 LARGE-SCALE PRETRAINING DATASET

Our pre-training dataset collection comprises 35 datasets, encompassing a total of 760K trajectories and 33M frames. Table 8 provides a comprehensive list of our pre-training datasets along with their respective sampling weights. The number of trajectories and the sampling weights can be automatically adjusted during dataset assembly. Following the prior data preprocessing approach (Kim et al., 2024), we reformulate the pre-training datasets to emphasize end-effector sequence control, ensuring alignment with the specific requirements of our model training. Due to inherent differences among datasets, only single 2D observations are used during pre-training. However, during fine-tuning, HybridVLA can accommodate both single- and multi-view observations depending on the

Table 8: The dataset name and sampling weight used in our mixed large-scale pretraining dataset.

| Training Dataset Mixture | |
|---|---:|
| Fractal (Brohan et al., 2022) | 9.1% |
| Kuka (Kalashnikov et al., 2018) | 27.8% |
| Bridge(Ebert et al., 2021; Walke et al., 2023) | 4.1% |
| Taco Play (Rosete-Beas et al., 2022; Mees et al., 2023) | 2.1% |
| Jaco Play (Dass et al., 2023) | 0.3% |
| Berkeley Cable Routing (Luo et al., 2023) | 0.2% |
| Roboturk (Mandlekar et al., 2018) | 1.7% |
| Viola (Zhu et al., 2023b) | 0.7% |
| Berkeley Autolab UR5 (Chen et al.) | 0.9% |
| Toto (Zhou et al., 2023) | 1.5% |
| Language Table (Lynch et al., 2023) | 3.1% |
| Stanford Hydra Dataset (Belkhale et al., 2023) | 3.2% |
| Austin Buds Dataset (Zhu et al., 2022) | 0.2% |
| NYU Franka Play Dataset (Cui et al., 2022) | 0.6% |
| Furniture Bench Dataset (Heo et al., 2023) | 1.8% |
| UCSD Kitchen Dataset (Yan et al., 2023) | <0.1% |
| Austin Sailor Dataset (Nasiriany et al., 2022) | 1.6% |
| Austin Sirius Dataset (Liu et al., 2023) | 1.2% |
| DLR EDAN Shared Control (Quere et al., 2020) | <0.1% |
| IAMLab CMU Pickup Insert (Saxena et al., 2023) | 0.7% |
| UTAustin Mutex (Shah et al., 2023) | 1.6% |
| Berkeley Fanuc Manipulation (Zhu et al., 2023a) | 0.6% |
| CMU Stretch (Mendonca et al., 2023) | 0.1% |
| BC-Z (Jang et al., 2022) | 5.4% |
| FMB Dataset (Luo et al., 2024) | 5.0% |
| DobbE (Shafiullah et al., 2023) | 1.0% |
| DROID (Khazatsky et al., 2024) | 7.2% |
| Stanford Kuka Dataset (Lee et al., 2019) | 0.1% |
| Stanford Robocook Dataset (Shi et al., 2023) | 0.1% |
| Maniskill (Gu et al., 2023) | 6.3% |
| Berkeley RPT (Radosavovic et al., 2023) | 0.1% |
| QUT Dexterous Manipulation (Ceola et al., 2023) | 0.1% |
| RoboSet (Kumar et al., 2023) | 1.8% |
| BridgeData V2 (Walke et al., 2023) | 4.7% |
| RoboMind (Wu et al., 2024b) | 5.2% |

task requirements. For instance, AgileX dual-arm robot tasks require three viewpoints, an ego view and two wrist camera views, to capture a comprehensive observation of the object while mitigating occlusions caused by the robot arm. HybridVLA processes multi-view images using a shared vision encoder and then concatenates the visual feature along the token dimension. Notably, the difference in the number of images used during pre-training and fine-tuning does not impact manipulation performance in downstream tasks.

## B.2 SELF-COLLECTED REAL-WORLD DATASET

The experimental assets and environments for the single-arm and dual-arm setups are shown in Figure 4 (a) and (b), respectively. For the single-arm setup, a 3D-printed UMI gripper (Chi et al., 2024) is attached to the Franka robot and is used across all baselines. We utilize RealSense 435 and RealSense 515 cameras to capture both wrist and front views. For the dual-arm setup, two Orbbec DABAI cameras are used to capture the left and right wrist views, while a RealSense 515 is mounted overhead to capture a static third-person view. We provide a detailed explanation of the real-world tasks and their success conditions. We begin by describing the single-arm tasks:

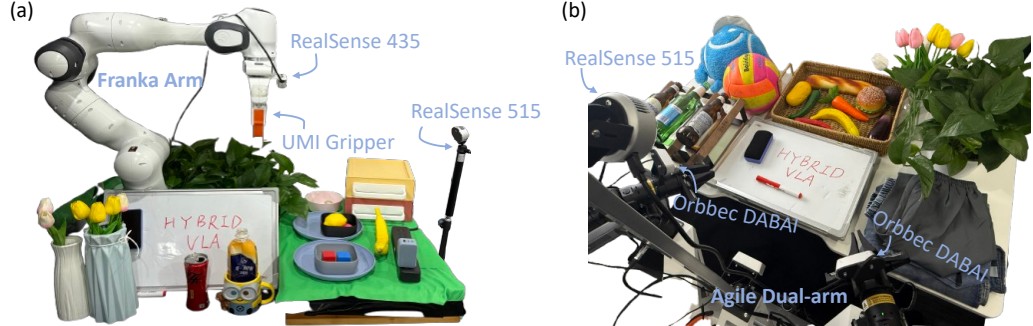

Figure 4: **Real-World Assets and Experimental Settings.** We provide visualizations of the assets used and the settings for single-arm FR3 robot tasks and dual-arm AgileX robot tasks, respectively.

*1. Pick and place.* This task requires the robot to pick up a specifically colored block based on a language description and place it in a specifically colored bowl.

*2. Unplug charger.* The robot needs to grasp the charger at an optimal position and rotation, and then lift it to a certain height without slipping.

*3. Pour water.* The robot needs to first pick the bottle, then rotate it to a position slightly above the cup, and tilt it to perform the pouring action. The task is deemed successful only if the bottle opening is correctly aligned with the cup.

*4. Wipe blackboard.* The robot needs to first grasp an eraser and then use it to remove the red markings from a blackboard placed on the tabletop. The red markings are drawn on an unfixed region, and the task is considered successful only if they are completely erased.

*5. Open drawer and place inside.* The robot needs to open the top drawer, pick up the required objects based on the language description, place them in the opened drawer, and then close it. This task consists of four sequential sub-tasks: *open drawer*, *pick object*, *place object*, and *close drawer*. The task is considered complete once all sub-tasks have been successfully executed.

We then describe the details of dual-arm tasks:

*1. Pick and place.* The robot must use both its left and right arms to pick up two objects based on the language description and place them in the container.

*2. Lift ball and place.* Both the left and right arms must simultaneously make contact with the ball, which is secured between the two grippers. The arms coordinate their movements to transport the ball to the container while ensuring it does not slip. This task highly tests the model's dual-arm coordination capabilities.

*3. Place bottles at rack.* The left and right robot arms need to grasp the bottles placed on their respective sides and rotate them to position them parallel to the rack.

*4. Wipe blackboard.* Unlike the single-arm setting, the dual-arm setting requires one arm to hold the whiteboard while the other picks up the eraser and wipes off the red marker.

*5. Fold shorts:* This task requires folding a pair of shorts, involving two sequential steps. First, one pant leg is folded over the other to align them. Then, the pants are folded in half from top to bottom. Throughout the process, both arms must coordinate their movements. For example, in the first step, the left arm holds the bottom of the pant leg while the right arm grips the upper part, working together to complete the folding.

## C  ADDITIONAL QUANTITATIVE RESULTS

### C.1  ADDITIONAL SIMULATION EXPERIMENTS

To further investigate the generalization capability of HybridVLA, we conduct experiments in the SimplerEnv (Li et al., 2024c) variant aggregation setting using the Google robot, which poses

Table 9: **Evaluation results on SimperEnv.** We evaluate our models in the variant aggregation setting of the Google Robot benchmark, where the number of test trials per scene follows the official protocol. All models are finetuned on the Fractal dataset. Bold indicates the highest score.

| Models | Pick Coke Can | Move Near | Open/Close Drawer | Open Top Drawer and Place | Mean S.R. ↑ |
|---|---|---|---|---|---|
| $\pi_0$ (2.6B) | 0.72 | 0.50 | 0.34 | 0.38 | 0.49 |
| HybridVLA (7B) | **0.84** | **0.64** | **0.40** | **0.48** | **0.59** |

significant challenges for evaluating a model's generalization to unseen configurations. Since the pretraining of $\pi_0$ does not include the Fractal dataset (Brohan et al., 2022) as a subset, unlike HybridVLA, we ensure a fair comparison by initializing both models with their respective pretrained checkpoints and finetuning them for 5 epochs on the same Fractal dataset. As shown in Table 9, HybridVLA consistently outperforms $\pi_0$ across 4 tasks. In particular, on tasks that demand strong scene understanding, such as *Open Top Drawer and Place*, HybridVLA achieves up to a 10% higher success rate.

## C.2    ADDITIONAL ABLATION STUDY

**The impact of confidence threshold in collaborative action ensemble.** The proposed collaborative ensemble strategy determines whether to use the action predicted by diffusion alone or the averaged output of both diffusion and autoregressive generations, guided by a mean confidence threshold derived from the autoregressive action token. In this experiment, we investigate the optimal confidence threshold required to ensure the accuracy of autoregressive actions and enhance the overall precision of the ensemble-generated action. Specifically, we evaluated HybridVLA on ten RLBench tasks, varying the confidence threshold from 0.90 to 0.98. The main results are presented in Table 10. We find that when the confidence threshold drops below 0.94, autoregressive predictions become unreliable, leading to a slight degradation in the performance of the ensemble action. Conversely, when the threshold reaches 0.98, the number of valid autoregressive actions becomes too limited, causing the performance of the ensemble action to closely match that of the diffusion-predicted action. Empirically, we conclude that setting the threshold to 0.96 ensures a stable action ensemble.

Table 10: **Confidence threshold.** We explore the impact of different confidence thresholds on the performance of ensemble actions. The model used for testing is HybridVLA (7B).

| Threshold | 0.90 | 0.92 | 0.94 | **0.96** | 0.98 |
|---|---|---|---|---|---|
| Mean S.R. ↑ | 0.68 | 0.72 | 0.75 | **0.78** | 0.74 |

**The impact of diffusion-based KV cache in inference speed.** As described in Section 3.3, we adopt the diffusion-based KV cache to eliminate redundant computations and improve inference speed. In this experiment, we examine the extent to which this mechanism accelerates inference. With the diffusion-based KV cache enabled (Table 2 of the main paper), HybridVLA-dif achieves an average success rate of 72% across 10 simulation tasks with an inference speed of 9.4 Hz. Removing it results in a similar average success rate but reduces the inference speed to 5.0 Hz. Although the KV cache has typically been used in previous autoregressive VLA methods (Kim et al., 2024; Li et al., 2024b), we are the first to integrate it into an LLM's diffusion-based action generation.

**The impact of denoising steps.** Figure 5 illustrates the relationship between manipulation performance and the number of denoising steps for HybridVLA-dif across ten RLBench tasks. Consistent with the findings of previous work (Bjorck et al., 2025; Liu et al., 2024b), we reduced the number of DDIM denoising steps of inference from 20 to 2 without observing a significant degradation in manipulation performance. To balance inference speed and accuracy, we set the diffusion denoising steps to 4 in our final implementation.

**The impact of weights between diffusion and AR losses.** As shown in Table 11, we conducted a detailed ablation study to examine how the dynamic weighting between the two losses influences manipulation success rates across the 10 simulation tasks. Except for adjusting the loss ratios, all other training settings remain identical to those used in the main paper. Since our model is pretrained on large-scale robotic datasets, the initial values of the two losses are similar. First, we observe

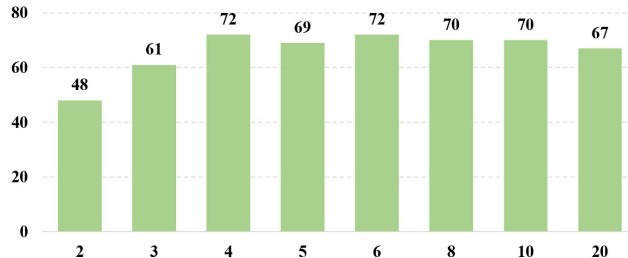

Figure 5: **The impact of denoising steps**, where the x-axis and y-axis represent the denoising steps and manipulation success rate.

that a ratio between AR and Diff models slightly above 1:1 yields a relatively stable average task success rate of approximately 0.78 to 0.80. When the ratio falls below 1:1, the performance becomes comparatively poorer. Additionally, we find that maintaining a ratio slightly above 1:1 leads to a marginally faster convergence speed during model training.

Table 11: Task success rates under different ratios of AR and diffusion losses.

| $\mathcal{L}_{AR} : \mathcal{L}_{Dif}$ | 10:1 | 5:1 | 2:1 | 1:1 | 1:2 | 1:5 | 1:10 |
|---|---|---|---|---|---|---|---|
| Mean S.R. ↑ | 0.79 | 0.80 | 0.78 | 0.78 | 0.75 | 0.77 | 0.75 |

**The impact of different temperature hyperparameters.** We added a sensitivity analysis for the LLM temperature hyperparameter across 10 simulation tasks. It is worth noting that some autoregressive VLA models do not explicitly set the temperature and instead directly select the token with the highest probability (Kim et al., 2024). Our experiments in the main paper also follow this setting. As shown in Table 12, we observe that when the temperature is less than or equal to 1, the manipulation success rate remains consistent. However, when the temperature exceeds 1, the action predictions become noticeably unstable. We observe that the robot arm may performs anomalous steps during closed-loop control, ultimately leading to a degradation in accuracy. The results demonstrate that, for robotic tasks, stability in action generation is far more important than output diversity. Therefore, it is reasonable to either adopt the OpenVLA strategy or use a relatively small temperature hyperparameter.

Table 12: Task success rates under different temperature settings.

| Temperature | no sample | 0.1 | 0.2 | 0.5 | 1.0 | 1.5 | 2.0 |
|---|---|---|---|---|---|---|---|
| HybridVLA | 0.78 | | 0.78 | 0.77 | 0.78 | 0.76 | 0.71 | 0.64 |

## C.3 ADDITIONAL GENERALIZATION EXPERIMENTS

To further investigate the spatial generalization capability of HybridVLA in real-world settings, we design a more stringent positional generalization benchmark for the pick-and-place task. In this experiment, we re-collected a dataset of 100 demonstrations with a precisely defined training distribution. The tabletop is divided into two non-overlapping spatial regions, from which object locations for the training and test sets are independently sampled. Figure 6 shows the visualization of the two regions. This setup enforces a clear positional distribution shift and provides a more challenging measure of generalization.

We evaluate HybridVLA(7B) and CogACT with 20 rollout episodes on the pick-and-place task. As shown in Figure 6, HybridVLA maintains strong performance despite the strict spatial separation between training and testing regions. The results demonstrate that the model is able to transfer learned manipulation behaviors to novel object locations outside the training distribution, highlighting its robustness in real-world positional generalization.

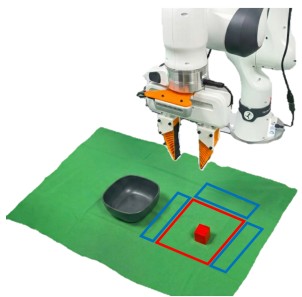

Figure 6: **Positional generalization visualization and results on real-world pick-and-place task.** The figure left shows the non-overlapping regions of training and testing, the red box refers to the training region, and the blue boxes are testing regions. The table below shows the success rate of the generalization experiments.

| Setting | HybridVLA | CogACT |
|---|---|---|
| Original region | 0.85 | 0.75 |
| Unseen position region | 0.75 (-11.8%) | 0.50 (-33.3%) |

## C.4 ADDITIONAL MOTIVATION EXPERIMENTS

We conducted additional experiments to further validate our motivation of leveraging an internet-scale pretrained LLM backbone as an action expert and combining the strengths of diffusion and autoregressive action generation, demonstrating clear advantages over using a separate diffusion head. Specifically, we constructed two variations of HybridVLA across 10 simulation tasks:

Variation 1: We append a Transformer-based diffusion head to HybridVLA and initialize it using the pretrained weights of the last two layers of the LLM backbone. To ensure consistency with prior diffusion-based VLM policies, we follow the token-processing scheme used in $\pi_{0.5}$: the diffusion head conditions only on the visual observation tokens and question tokens, and noise is injected at the action head for diffusion modeling. Following the $\pi_{0.5}$ (Intelligence et al., 2025) training paradigm, the LLM's AR branch predicts discrete tokens, whereas the additional diffusion head outputs continuous actions.

Variation 2: Under this setup, we essentially replace $\pi_0$'s VLM with HybridVLA's backbone, without using any part of HybridVLA's LLM-based AR or diffusion generation pathways. We append the same Transformer diffusion head, allowing the model to rely solely on the diffusion loss. Similar to $\pi_0$ Black et al. (2024) and consistent with Variation 1, the diffusion head is still conditioned solely on visual observation tokens and question tokens.

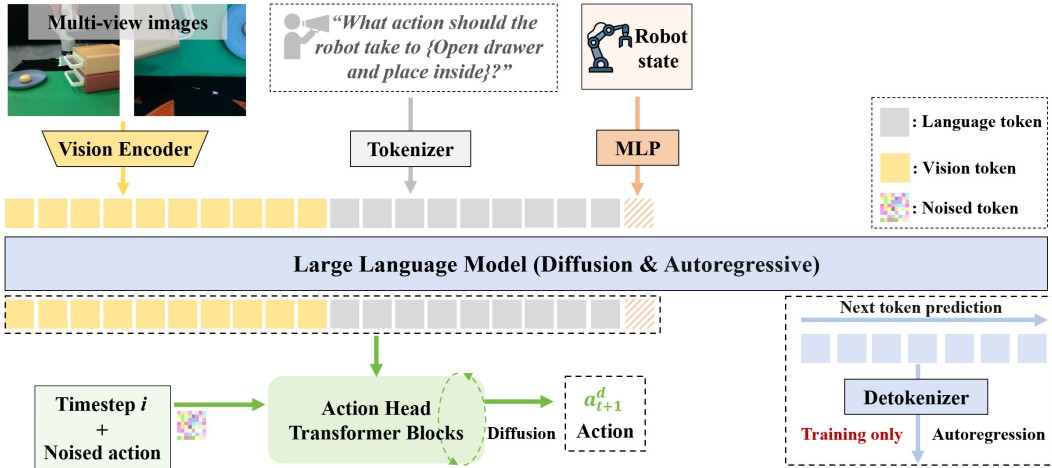

Figure 7: **The model architectures of variation1.** The transformer-based diffusion head is attached to HybridVLA.

Figure 7 and 8 shows schematic diagrams of the two variant model structures. We show the quantiative results on the 10 tasks of these different models in Table 13. First, compared with HybridVLA-dif, both Variation 1 and Variation 2 show a noticeable performance degradation. These results support our motivation and highlight the advantage of our approach: embedding the Markovian denoising steps of diffusion into the next-token prediction process allows each denoising step to function as a reasoning iteration within the LLM backbone, thereby fully leveraging the internet-scale

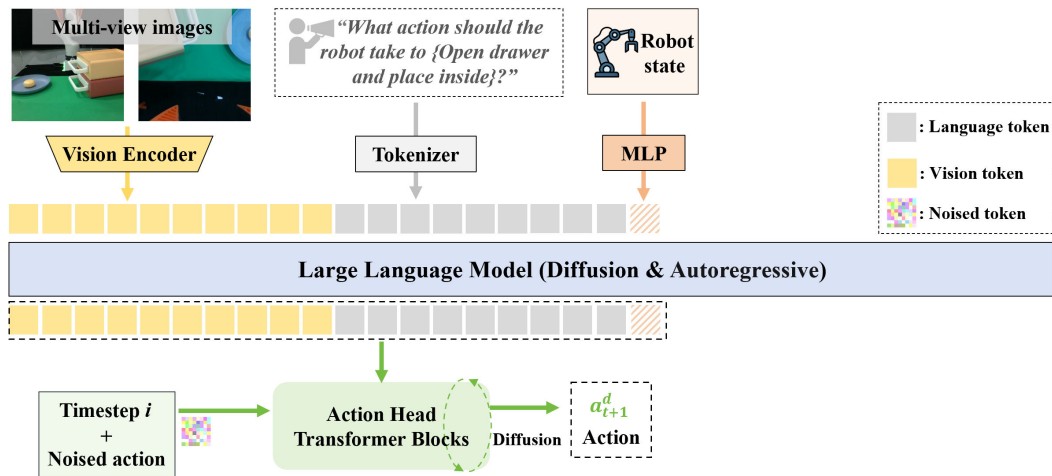

Figure 8: **The model architectures of variation2.** The same transformer diffusion head is attached but AR action generation from HybridVLA is disabled.

pretrained knowledge of the LLM. Simply attaching a diffusion head and loading pretrained weights is insufficient, because VLMs are pretrained using the full 32-layer Transformer architecture for forward feature propagation. Therefore, preserving the LLM's inherent contextual modeling paradiam is essential for achieving more robust diffusion-based action generation.

Next, when comparing HybridVLA-dif with the baseline Ex3, and similarly comparing Variation 1 with Variation 2, we observe that introducing AR generation consistently improves diffusion-based action accuracy across all variants. These results demonstrate that the AR branch inherits the VLM's pretrained generation paradigm, which enables it to learn from demonstrations more efficiently. Meanwhile, this finding also reinforces our motivation and the strength of our method: the two action paradigms can mutually reinforce each other and thereby enhance overall action robustness.

Table 13: Task success rates of HybridVLA-dif and variants.

| Method | HybridVLA-dif | Ex3 (Ablation Table 3) | Variation 1 | Variation 2 |
|---|---|---|---|---|
| Mean S.R. | 0.72 | 0.65 | 0.67 | 0.59 |

## D  ADDITIONAL VISUALIZATIONS

Figure 9 and Figure 10 illustrate keyframes of single-arm and dual-arm real-world execution processes. Notably, our Franka Research 3 (FR3) operates with controller version 5.6.0, libfranka version 0.13.3, Franka ROS version 0.10.0, and Ubuntu 20.04 with ROS Noetic. Under these software settings, the FR3 remains in *green* light execution mode with the FCI switch set to 'on'.

These tasks demonstrate HybridVLA's capability in accurately predicting position and rotation, as well as determining the precise timing for changing the gripper's open state. Additionally, the dual-arm tasks highlight HybridVLA's ability to coordinate both robotic arms, enabling it to complete tasks beyond the capability of a single arm, such as transporting a ball to a container. Notably, the single-arm task 'open drawer and place' and the dual-arm tasks 'wipe whiteboard' and 'fold shorts' are long-horizon tasks that involve at least three atomic sub-tasks. These results further confirm that HybridVLA can reliably predict long-horizon actions, demonstrating its capability to complete extended tasks.

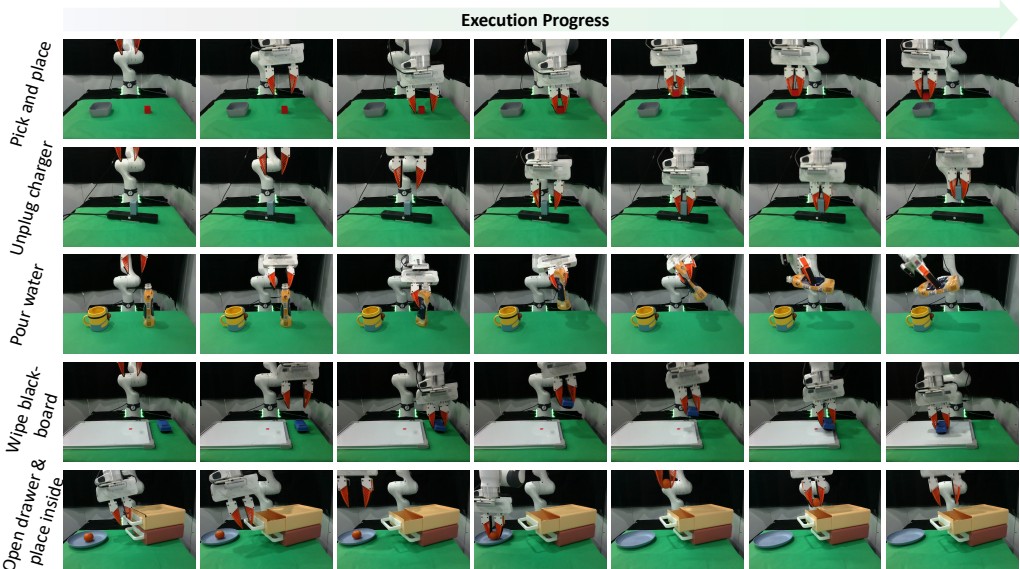

Figure 9: **Single-arm Execution Visualization**. We visualize key frames of the agent's execution process from the front perspective.

# E    FAILURE CASE ANALYSIS.

Through extensive real-world experiments, we identify three primary failure categories that impact the performance of HybridVLA. The first category, **rotational prediction deviations**, is particularly evident in tasks requiring precise rotation control, such as *Pour water* and *Place bottle at rack*. These failures include accumulated errors in multi-step rotational movements and incorrect rotation angles when interacting with target objects. The second category pertains to pose predictions that exceed the robot's **degree of freedom limits**. The model sometimes predicts poses beyond the mechanical constraints of the Fr3 arm or AgileX dual-arm robot, generates target positions that fall outside the workspace boundaries, or produces kinematically infeasible configurations during complex transitions. The third category involves failures in **dual-arm coordination**, where both arms must collaborate to complete a task. Since the model predicts each arm's actions based on the current object state, any interaction by one arm can alter the object's state, potentially invalidating the previously predicted action of the other arm.

# F    THE USE OF LARGE LANGUAGE MODELS (LLMS).

The research ideation of this paper did not involve any assistance from LLMs. However, during the writing process, we employed GPT (Achiam et al., 2023) to check grammar and refine word choice, aiming to ensure rigor in the manuscript. In addition, when constructing the language-based task plan data, we utilized LLMs by first performing manual annotations and subsequently applying GPT for automated augmentation and validation.

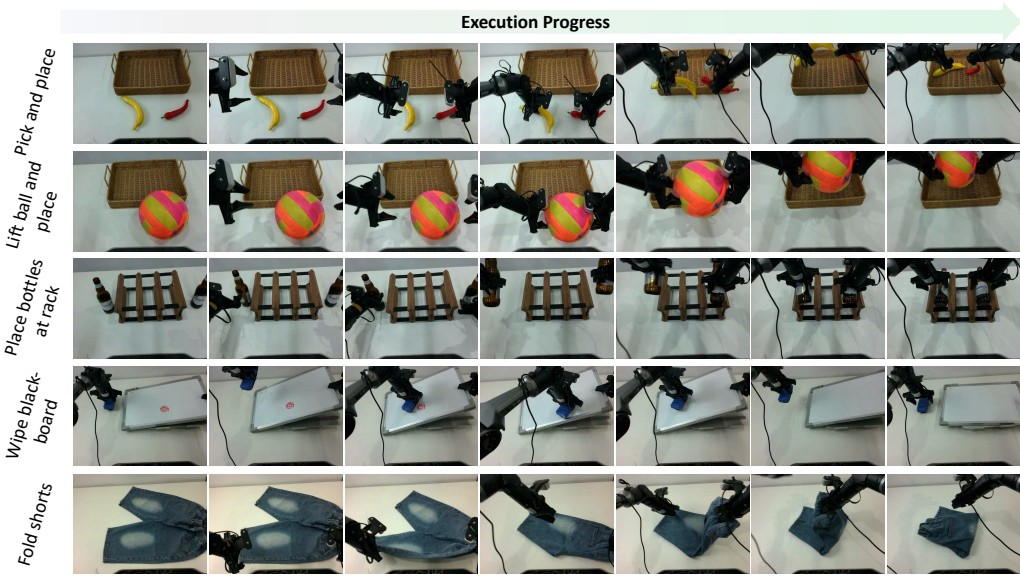

Figure 10: **Dual-arm Execution Visualization**. We visualize key frames of the agent's execution process from a static exterior view.

