# OpenReview forum: "HybridVLA: Collaborative Diffusion and Autoregression in a Unified Vision-Language-Action Model"
_ICLR.cc/2026/Conference — ICLR 2026 Poster_

### Official Review · Reviewer_4X5g · 2025-10-26

**Soundness:** 2
**Presentation:** 3
**Contribution:** 2
**Rating:** 2
**Confidence:** 5

**Summary:**

This paper proposes HybridVLA, a policy architecture that injects diffusion into autoregressive action generation in VLM inference. The proposed method is evaluated on RLBench and shows promising results against other VLA variants. The method is also evaluated on real world single arm / dual arm scenarios.

**Strengths:**

1. The paper overall is easy to follow.
2. This paper aims to design novel policy architecture that is based on well-established techniques, i.e., diffusion, autoregression, which is a valuable problem and promising direction.
3. The proposed method is evaluated on different tasks with single-arm and dual-arm real robots.

**Weaknesses:**

My major concerns are two: unclear motivation and limited novelty.

- Motivation: The proposed techniques are driven by the motivation to mix diffusion and autoregressive together. Instead of having a diffusion head at the end of the autoregression, we can incorporate diffusion into the autoregression steps. However, I seem to be unable to find a compelling argument or motivating example in this paper, which can convince me why this new formulation can be superior to having a diffusion head only (which is much simpler, and also effective as proven in many works).

- Novelty: The paper proposed to (1) train the HybridVLA with teacher forcing loss and denoising loss combine, and (2) ensemble the autoregressive and diffusion action output. Both techniques are engineering-driven modifications, lack novelty or general application potential.

- Missing critical references: CDP[1] and ARP[2].
   - Causal Diffusion Policy (CDP) studies the same problem and proposes a similar architecture, yet it was not compared or mentioned in this paper (correct me if I am wrong). The authors need to address this comparison.
   - Autoregressive Policy (ARP) also supports action-chunk prediction within the autoregressive steps, like the proposed architecture in this paper, which also works with teacher-forcing training (without a collaborative training stage). Moreover, ARP is also evaluated on RLBench (using the same demonstrations for training data) and achieves 84% success rates on 18 tasks, despite being a smaller model. The authors are also recommended to compare with other policies (like 3D diffusion policy) on RLBench.

[1] CDP: Towards Robust Autoregressive Visuomotor Policy Learning via Causal Diffusion (CoRL 2025)

[2] Autoregressive Action Sequence Learning for Robotic Manipulation (RAL 2025)

- The figure in Table 1 is very small, the robot state token and the autoregressive token almost have the same color (when printed), which makes it very difficult to understand.

**Questions:**

(please see weakness section)

---

> ### Author Response · Authors · 2025-11-23
> **Reply to W1**
>
> First of all, thank you for recognizing that our proposed HybridVLA “designs a novel policy architecture based on well-established techniques” and is “easy to follow.” We sincerely appreciate your thoughtful comments and questions, which have further strengthened our paper, and we address them in detail below.
>
>
>
> ## [W1.] Additional Motivation Experiments
>
> Following your constructive suggestion, we conducted additional experiments to further validate our **motivation of leveraging an internet-scale pretrained LLM backbone as an action expert and combining the strengths of diffusion and autoregressive (AR) action generation**, demonstrating clear advantages over using a separate diffusion head.
>
>
> Specifically, we constructed two variations of HybridVLA across 10 simulation tasks:
>
> **Variation 1:** We append a Transformer-based diffusion head to HybridVLA and initialize it using the pretrained weights of the last two layers of the LLM backbone. This configuration enables: (1) autoregressive action generation through the HybridVLA outputs, and (2) diffusion-based action generation via the duplicated Transformer head. All other settings remain identical to our method.
>
> **Variation 2:** We append the same Transformer diffusion head but disable AR action generation from HybridVLA, allowing the model to rely solely on the diffusion loss.
>
> As shown in the table below, we compare these two variants with HybridVLA-dif (jointly trained using our proposed collaborative training recipe) and with the baseline Ex3 (ablation Table 3, which relies solely on the diffusion loss). Note that all methods rely exclusively on diffusion-based actions during testing.
>
> First, compared with HybridVLA-dif, both Variation 1 and Variation 2 show a noticeable performance degradation. These results support our motivation and highlight the advantage of our approach: embedding the Markovian denoising steps of diffusion into the next-token prediction process allows each denoising step to function as a reasoning iteration within the LLM backbone, thereby fully leveraging the internet-scale pretrained knowledge of the LLM. Simply attaching a diffusion head and loading pretrained weights is insufficient, because VLMs are pretrained using the full 32-layer Transformer architecture for forward feature propagation. Therefore, preserving the LLM’s inherent contextual modeling paradiam is essential for achieving more robust diffusion-based action generation.
>
> Next, when comparing HybridVLA-dif with the baseline Ex3, and similarly comparing Variation 1 with Variation 2, we observe that introducing AR generation consistently improves diffusion-based action accuracy across all variants. These results demonstrate that the AR branch inherits the VLM’s pretrained generation paradigm, which enables it to learn from demonstrations more efficiently. Meanwhile, this finding also reinforces our motivation and the strength of our method: the two action paradigms can mutually reinforce each other and thereby enhance overall action robustness.
>
> |  | HybridVLA-dif |  Ex3 (ablation Table 3) | Variation 1 | Variation 2 |
> | --- | --- | --- | --- | --- |
> | Mean S.R. | 0.72 | 0.65 | 0.67 | 0.59 |
>
>
> Finally, as stated in Lines 151 to 155 of the original submission (Lines 157 to 161 of the revised submission), we also explain our motivation for unifying the two generation paradigms. Diffusion and autoregressive methods offer complementary strengths, providing fine-grained continuous control as well as efficient demonstration learning and semantic understanding. In Appendix A.1, we further validate this motivation through quantitative justification across fine-grained manipulation, dynamic-object scenarios, unseen-object settings, and language-instruction generalization. Following your constructive suggestion,  we will added the above justifications to the revised submission to further validate the motivation and the advantages of our proposed method.

---

> ### Author Response · Authors · 2025-11-23
> **Reply to W2**
>
> ## [W2.] Additional Novelty Clarification
> **First**, thank you for your detailed comments. We would like to clarify that "training HybridVLA with teacher forcing loss and denoising loss combined" describes only our hybrid objectives, but it is not the technical contribution of our paper. Our primary contribution lies in proposing a unified token-sequence formulation that embed diffusion and AR action generation into a single model, enabling them to mutually reinforce each other. **Following your insightful comments, we present two sets of evidence demonstrating that the effectiveness and contribution of HybridVLA do not depend solely on the combination of the two losses.**
>
> **1) Hybrid loss under different token sequence formulations.**
> To show that the effectiveness originates from our token-sequence design, we evaluate several alternative formulations, as shown in Table 1 of the submission. Both Type 2 and Type 3 incorporate two losses, but use different token-sequence designs. We compile a new table below that includes Type 2, Type 3, HybridVLA, and the baseline Ex3 from ablation Table 3 (which relies solely on the diffusion loss). Compared with baseline Ex3, Type 2 and Type 3 do not show any performance improvements despite using the hybrid loss, **demonstrating that the hybrid loss must be paired with the correct token-sequence formulation in order to enable mutual reinforcement between the two paradigms.** Together with the results of Variation 1 in our W1 response, which **show that attaching a diffusion head with the hybrid loss still fails to yield stable action generation**, these findings confirm the importance of our proposed unified token sequence.
> ||Type 2|Type 3|Ex3 (Table 3 in the main paper)|HybridVLA|
> |-|-|-|-|-|
> |Mean S.R.|0.56|0.65|0.65|0.78|
>
> **2) Hybrid loss with a modified causal attention mask.**
> We conduct an additional experiment on our proposed token sequence by modifying the causal attention mask. Specifically, after diffusion denoising, we mask out the diffusion tokens so that they cannot serve as conditioning signals for AR action generation, thereby removing the continuous contact prior while keeping the hybrid loss unchanged. We refer to this variant as **HybridVLA-mask**. As shown in the table below, HybridVLA-mask exhibits an 5% performance drop compared to HybridVLA, despite both models using the same token sequence and hybrid objectives. **These results further support our main contribution: it is the unified token sequence, together with the properly designed causal latent conditioning, that strengthens action generation. The resulting improvements cannot be soly explained by hybrid loss.**
> ||HybridVLA|HybridVLA-mask|
> |-|-|-|
> |Mean S.R.|0.78|0.73|
>
> **Second**, we would like to clarify that ensembling the autoregressive and diffusion action outputs depends on our unified model design and our key observation that the two generation paradigms exhibit complementary strengths across different tasks. Specifically, as shown in Appendix A.1 and Figure 3 of the submission, diffusion-based generation excels at producing fine-grained, temporally consistent actions. In contrast, autoregressive action generation inherits the pretrained generation paradigm of VLMs, enabling more efficient learning from demonstrations (Pertsch et al., 2025; Intelligence et al., 2025) and demonstrating strong robustness in language understanding and generalization to novel objects. **Therefore, our contribution lies primarily in identifying these complementary properties of the two paradigms. Once the model is equipped with both robust AR and diffusion action generation, a simple yet effective ensembling strategy naturally leads to more stable manipulation performance.**
>
> **Finally**, our method is _not_ an engineering-driven modification. To the best of our knowledge, we are the first in the VLA domain to explore unifying the diffusion and autoregressive generation paradigms within a single LLM, thereby combining their complementary strengths for robotic action generation. As demonstrated in the experiments above, simply merging two losses does _not_ yield any performance gain. **This shows that our method is not a trivial engineering tweak, but instead relies on the carefully designed unified token sequence and causal conditioning mechanism.**  Beyond accuracy improvements, we further justify our method in Appendix A.2 using a principal component analysis of the latent distribution, showing that it not only strengthens feature representations but also implicitly regularizes the deeper latent space. Moreover, HybridVLA can aslo achieve unified planning and action. As shown in Lines 364–369 of the submission, we modify the AR outputs from discrete actions to language-based sub-goal plans, while the diffusion branch continues to generate actions. **HybridVLA achieves a strong 74% mean success rate while performing sub-goal planning, demonstrating its general applicability and extensibility.**

---

> ### Author Response · Authors · 2025-11-23
> **Reply to W3 and W4**
>
> ## [W3.] Additional discussion and comparison with related works.
>
> Thank you for your constructive comments. We will add the discussion and comparison with the baseline you mentioned, and have updated related work and Table 2 in the submission paper accordingly, with the revised sections highlighted in blue.
>
> **1) Regarding Causal Diffusion Policy (CDP),** this method introduces an autoregressive Transformer-based action generation mechanism. By incorporating historical action chunks as part of the input and designing specialized attention masks, CDP provides long-horizon conditioning for future action prediction. In contrast, HybridVLA focuses on unifying diffusion-based and autoregressive action generation paradigms within a single model and a unified token sequence. This design enables the model to fully exploit the internet-scale pretrained knowledge of the LLM while leveraging the complementary strengths of the two generation paradigms. Through joint optimization, the two paradigms mutually reinforce each other, thereby enhancing manipulation stability. **In summary,** **although CDP and HybridVLA both adopt Transformer architecture, their motivations and problem formulations differ**. CDP aims to improve long-horizon action prediction by integrating historical action information, whereas HybridVLA is the first to demonstrate how to integrate and mutually enhance two complementary action generation within a unified framework. Since CDP has not released evaluation code for RLBench, we will include the corresponding citation and analysis in the revised submission.
>
> **2) Regarding Autoregressive Policy (ARP)**, it is built upon an autoregressive Transformer architecture and predicts task-specific action chunks, aiming to balance action accuracy with generation efficiency.  The core motivations of the two approaches are different. ARP focuses on designing efficient action chunks to achieve fast and accurate inference, whereas HybridVLA aims to unify and mutually enhance the two generation paradigms, forming a unified VLA model in which the two paradigms reinforce each other. In addition, once HybridVLA is equipped with both generation paradigms, the AR branch can also generate language-based sub-goal plans, as shown in Lines 364–369 of the submission. Following your suggestion, we compare ARP with HybridVLA on 10 RLBench tasks, strictly following the official experimental settings. Note that for ARP, we report two versions: one using four camera views (**ARP4**), and another using only the front-view camera (**ARP1**). The latter matches the camera configuration used by all baselines in our paper. As shown in the table below, HybridVLA consistently outperforms both ARP4 and ARP1. These results highlight two key aspects: (1) our innovative use of an LLM as the action expert allows HybridVLA to better inherit internet-scale pretrained knowledge, providing stronger model capacity; and (2) our unified token sequence and unified representation effectively capture the complementary strengths of the two generation paradigms, yielding more robust action predictions.
>
> Since 3D Diffusion Policy does not include a text encoder for interpreting different task instructions, its performance in multi-task settings is unsatisfactory. We sincerely appreciate your valuable suggestions on these additional comparison experiments, and we will incorporate all of them into the main paper accordingly.
> ||Close box| Close laptop | Toilet seat down | Sweep to dustpan | Close fridge | Phone on base | Take umbrella out of stand |Take frame off hanger|Place wine at rack|Water plants|Mean|
> |:-:|:-:|:-:|:-:|:-:|:-:|:-:|:-:|:-:|:-:|:-:|:-:|
> |HybridVLA|95|95|100|90|100|55|60|70|60|55|78|
> |ARP4|60|90|100|90|95|45|50|40|70|40|68|
> |ARP1|35|60|75|80|70|30|40|25|35|20|47|
>
> **3) Finally, we would like to further clarify why we selected the baselines used in our paper.** Our main contribution is the design of a novel **VLA model** that **unifies autoregressive and diffusion action generation**, leverages their complementary strengths, and enables mutual reinforcement to improve action generation robustness. Therefore, we compare HybridVLA against **prior state-of-the-art VLA models** that adopt **different generation paradigms**, including ManipLLM (Li et al., 2024b), OpenVLA (Kim et al., 2024), OpenVLA-OFT (Kim et al., 2025), π0 (Black et al., 2024), and CogACT (Li et al., 2024a). In addition, we evaluate the generalization capability of our method across different simulation settings (e.g., SimplerEnv), as shown in Appendix C.1.
>
> ## [W4.] Update of Table 1 and Figure
> We sincerely apologize for the inconvenience caused by the figure colors during your reading. We overlooked the fact that, after printing, the robot-state tokens and autoregressive tokens appeared in similar colors. We have updated the Table 1 and Figure 2 in the revised submission PDF and have carefully verified that the token colors are now clearly distinguishable in print.

---

> ### Comment · Reviewer_4X5g · 2025-11-25
> **response**
>
> I really appreciate the efforts put in for new experiments. But it hasn't fully resolved my concerns yet. Let me elaborate below:
>
> 1. The motivation of using LLM and combining diffusion is too general, and it is not what I am looking for. I am looking for an explanation/argument of why placing autoregression after diffusion action prediction can actually enhance action prediction, and why not just placing autoregression before diffusion action prediction.
>
> For example, if a robot model predicts: "the action is [vector] to reach the cabinet", where the [vector] is predicted from diffusion. Then, 1) why I still need to predict the "to reach the cabinet", 2) and why not I just predict a sequence "to reach the cabinet, the action is [vector]" (reorganize the sequence so the action vector is placed at the end.
>
> 2. The new experiment is clear but I wonder it may not be the most popular/standard way to use diffusion head, like the way in pi0/pi0.5.  The experiment that I am looking for is to cut off all remaining seqences after the diffusion prediction. Like comparing the results of generating "the action is [vector] to reach the cabinet" and "the action is [vector]". But not applying additional transformer layers on the tokens of "the action is ".
>
> Updated: The new table 1 seems provide some experiment results.
>    - Could you give some example sequences of what those hybrid sequences look like? The result looks good but I want more explanation of why this would work to convince myself.
>    - Could you try modifying Type-3 to "[VL][VL][VL][Auto][Auto][Auto][Cont Robot State][BOD][Diffusion][BOD]"?
>
> 3. I don't feel like categorizing a work as "VLA" is a good reason to waive the comparison against other policies developed before the "VLA" trend. Because it practically follow the same evaluation protocol as previous work did and does not outperform under 2.7B parameters (still much larger model than baseline).
>
> However, I do appreciate the efforts put into the work by the authors. I will increase the score to borderline, but I still believe the motivation and novelty are of serious concern.

---

> > ### Author Response · Authors · 2025-11-27
> > **Reply to Point 3**
> >
> > ### **[Point 3] More comparisons between VLAs and prior methods**
> > First, our choice of baselines is guided by the baselines commonly used in earlier VLA studies. At the same time, we fully agree with the perspective that VLA models should not be compared only within the “VLA” family, and that evaluations against other policies developed before the emergence of VLA frameworks remain important and valuable.
> >
> > To this end, as discussed in **our response [W3]**, we compared HybridVLA (7B, single-view) with ARP1 (single-view) and ARP4 (multi-views). Despite using only a single camera, HybridVLA demonstrates a clear advantage in multi-task success rates. In addition, under the single-view setting, we compare HybridVLA (2.7B) against ARP1. As shown in the table, HybridVLA (2.7B) achieves higher task success rates, further demonstrating its strong robustness in action generation.
> >
> > We adopt a single-view setup to make the model more practical and cost-effective. Nevertheless, we additionally compare a multi-view setting between HybridVLA (2.7B) and ARP4. The experimental results in the table below and indicate that incorporating richer visual observations yields consistent improvements for HybridVLA.
> > ||HybridVLA (single-view, 2.7B)|ARP1|HybridVLA (multi-views, 2.7B)|ARP4|
> > |-|-|-|-|-|
> > |Mean S.R.|0.67|0.47|0.80|0.68|
> >
> > **Notably, ARP, which adopts a transformer-based architecture, further explores the paradigm of action-chunking outputs, improving the accuracy and efficiency of AR action generation. This provides valuable insights for discrete action generation in VLA research, and offers meaningful guidance for enhancing both the capability and efficiency of HybridVLA in the future. We have cited ARP in the updated version of our paper and have articulated these connections in the revised version.**

---

> ### Author Response · Authors · 2025-11-27
> **Reply to Point 1**
>
> First of all, we are grateful to receive your follow-up discussion and the further clarification of your concerns. In the response below, we carefully address the new points you raised and have conducted additional validation following your constructive suggestions.
> ### **[Point 1] Analysis of the ordering between autoregression (AR) and diffuison (Dif)**
> **1) Detailed description of the token sequence**
>
> We believe that providing a clear description of the input token sequence can facilitate a better understanding of HybridVLA’s training strategy. Therefore, we first outline the token sequence used in our final design (Type 4 in Table 1, Figure 2), as shown in the box below:
> ```text
> <vision> + <language prompt> + <continuous robot state> + <BOD> + <diffusion noised actions> + <EOD> + <discrete action>
> ```
> For clarity, this sequence can also be expressed as:
> ```text
> [Vision][Text][Continuous robot state][BOD][Diffusion][EOD][AR-action][AR-action][AR-action]
>  ```
> The `<language prompt>` instruct the model in an interrogative form, for example: *“What action should the robot take to put the phone on the base?”* The `<continuous robot state>` and `<diffusion noised actions>` are obtained by feeding robot states and noised actions through an MLP to project them into the LLM embedding space. If the model controls a single-arm robot, the `<discrete action>` follow the representation described in lines 152–156 of the submission: “Δx, Δy, Δz, Roll, Pitch, Yaw, 0/1,” resulting in a 7-token action. For dual-arm control, the action simply doubles to 14 tokens. In this setting, the `<diffusion noised actions>` and `<discrete action>` are supervised during training using MSE loss and cross-entropy loss, respectively.
>
> **2) Analysis**
>
> We explain why we place the `<discrete action>` after the `<diffusion noised actions>` from two perspectives:
> - **Ground-truth leakage.**
> As described in lines 249–255 of our submission, due to the AR training nature of LLMs, both the “question prompt tokens’’ and the “answer tokens’’ are input to the LLM. Therefore, in our training, if the input token sequence is **[Vision][Text][Continuous robot state][AR-action][AR-action][AR-action][BOD][Diffusion][EOD]**, the diffusion tokens will inevitably attend to the AR action ground-truth ([AR-action] tokens). This results in action GT leakage, since all preceding tokens, including the [AR-action] tokens, serve as conditioning signals for diffusion modeling. As indicated by our experimental analysis (Type 3 & Type 4 in Table 1), this shortcut causes the diffusion model to rely on leaked action cues rather than learning to model actions on its own, which ultimately leads to degraded performance.
> - **Learning hierarchy.** To further analyze how the two ordering strategies affect the learned action representations, we describe the two settings separately. In the **AR → Dif** setup, the `<discrete action>` are placed before the `<diffusion noised actions>`. In the **Dif → AR** setup, the `<discrete action>` are placed after the `<diffusion noised actions>` instead. To evaluate the impact of these two designs, we conducted a principal component analysis (PCA) following the procedure described in Appendix A.2 of our submission. Specifically, we sampled several trajectories from the Pick and Place tasks and fed their corresponding scene images into the model. For each input, we extracted the diffusion-denoised actions and the AR action tokens, then projected their features into a 2D space using PCA. As shown in the table below, the **Dif → AR** design produces diffusion and AR features that form tighter intra-class clusters and exhibit larger inter-class separation for both Pick and Place actions. When `<discrete action>` is placed after `<diffusion noised actions>`, the reduced Intra/inter ratio indicates a more structured latent organization during next-token prediction. Diffusion branch learns a smooth and structured continuous action manifold, while the AR branch maximizes the conditional action probability and directly models the action distribution. Designing the token sequence as **Dif → AR** aligns with this optimization hierarchy: diffusion first shapes a smooth latent action space, and the AR branch then performs conditional reasoning and refinement on top of this structured representation. This suggests that the sequence order **Dif → AR** not only strengthens cross-action feature representations but also aligns more closely with the underlying action distribution, ultimately enabling more stable action generation.
>
> **Table A:**
> ||AR→Dif||Dif→AR||
> |-|-|-|-|-|
> ||AR action|Dif action|AR action|Dif action|
> |Intra–class distance|0.60|0.63|0.44|0.49|
> |Inter–class distance|8.1|5.6|10.8|8.7|
> |Intra/inter ratio|0.07|0.11|0.04|0.06|
>
> We sincerely appreciate your insightful guidance, which allowed us to clarify the motivation behind our token-sequence design more systematically. We will incorporate all of your suggestions into the revised version.

---

> ### Author Response · Authors · 2025-11-27
> **Reply to Point 1 and Point 2**
>
> **3) When HybridVLA autoregressive output sub-goal text (your raised example)**
>
> In **1) and 2) of Point 1**, we discussed and analyzed the advantages of placing the autoregressive outputs after the diffusion actions. As noted in lines 364–369 of the submission, the `<discrete action>` can be replaced with sub-goal text for more general-purpose settings. Importantly, once the autoregressive branch predicts sub-goal text rather than low-level action tokens, the issue of action ground-truth leakage no longer arises. Thanks to our designed simple yet effective marker tokens (`<BOD>`/`<EOD>`), HybridVLA can freely switch the ordering between autoregressive and diffusion predictions.
> Following your insightful suggestion, we compared three configurations: predicting sub-goal text before diffusion actions (HybridVLA-Lang-before), predicting sub-goal text after diffusion actions (HybridVLA-Lang-after), and using diffusion action modeling alone. As shown in the table below, placing the sub-goal text before or after diffusion leads to similar average success rates, both of which outperform the diffusion-only setting. This indicates that even when the autoregressive branch predicts high-level sub-goal descriptions that share only semantic intent (e.g., *“grasp the lid of the box”*) rather than modeling the same action distribution, the two paradigms still mutually reinforce each other and yield stronger overall performance.
>
> **Table B:**
> ||Mean S.R.|
> |-|-|
> |HybridVLA-Lang-before|0.73|
> |HybridVLA-Lang-after|0.74|
> |Ex3 in Table 3 of main paper (diffusion action alone)|0.65|
>
> ### **[Point 2] Additional explanations and experiments**
>
> **1) Diffusion head usage explanations**
>
> Thank you for your detailed comments. We apologize for not clearly explaining how the diffusion head is used in **Variation 1 and Variation 2 in our response [W1]**.
>
> - For **Variation 1**, we follow the token-processing scheme used in π₀.₅. The diffusion head receives the `<vision>` and the `<language prompt>` as conditioning information, and noised action is injected at the action head to perform diffusion modeling. In this setting, we simply replace π₀.₅’s VLM with HybridVLA’s backbone. The LLM’s AR branch predicts discrete actions, while an additional diffusion head outputs continuous actions.
>
> - For **Variation 2**, we adopt the π₀ training paradigm. In this setting, the input portion remains the same, but we use only the action head to output continuous actions.
>
> To make our reproduction setup clearer, we have added **Figure illustrations for both Variation 1 and Variation 2 in Appendix C.4** of the revised submission PDF, enabling easier inspection of the corresponding frameworks.
>
> **2) Additional experiments anlysis**
>
> Regarding your question on **comparing the results of generating 'the action is [vector] to reach the cabinet' and 'the action is [vector]'**, our interpretation is as follows:
>
> First, “the action is” should fall under the category of `<language prompt>`. These tokens serve solely as conditioning information and are **not** supervised. The token sequence “[vector]” corresponds to `<diffusion noised actions>`, which are trained using an MSE loss. The phrase “to reach the cabinet” belongs to the sub-goal text tokens, which are supervised using a cross-entropy loss.
>
> Under this interpretation, the experiment corresponding to **'the action is [vector] to reach the cabinet'** matches **HybridVLA-Lang-after**, described in **our response Point 1 Table B**. Meanwhile, **'the action is [vector]'** corresponds to **Ex3 in the main paper (Table 3)**, also referenced in **our response Point 1 Table B**.
>
> Based on the repoted results, our proposed unified token sequence not only avoids negative interference between the two paradigms but also enables clear mutual reinforcement. The findings further show that even when the AR branch predicts sub-goal text that expresses only high-level semantic intent (e.g., *“grasp the lid of the box”*) rather than modeling the same action distribution, the two paradigms still reinforce each other and yield superior performance.

---

> ### Author Response · Authors · 2025-11-27
> **Reply to updated suggestions**
>
> ### **[Updated 1] Example token sequences in Table 1**
> In **our respose Point 1**, we introduced the token sequence design ultimately adopted by HybridVLA, corresponding to **Type 4** in Table 1. Following your suggestion, we further describe the other token-sequence variations in Table 1 and analyze why Type 4 is the superior choice, as also discussed in lines 235–256 of the submission.
> - **Type 1** differs from Type 4 in that it uses `<discrete robot state>` instead of continuous ones. The discrete robot states share a similar representation format with `<discrete action>`. Our experiments show that `<continuous robot states>` are more compatible with our training paradigm and lead to more robust action learning.
>     ```text
>     <vision> + <language prompt> + <discrete robot state> + <BOD> + <diffusion noised actions> + <EOD> + <discrete action>
>      ```
> - **Type 2** removes the special tokens `<BOD>` and `<EOD>` used to mark the beginning and end of the diffusion modeling. Without these boundary indicators, the `<diffusion noised actions>` must learn not only the noise-prediction objective inherent to diffusion modeling but also the prediction of the first token of `<discrete action>`. This conflation of objectives complicates optimization and degrades learning stability.
>     ```text
>     <vision> + <language prompt> + <continuous robot state> + <diffusion noised actions> + <discrete action>
>      ```
> - **Type 3** swaps the prediction order of `<diffusion noised actions>` and `<discrete action>`. This design makes the GT action directly visible to the diffusion action modeling process as conditioning signals, resulting in GT leakage. As supported by our experimental findings, such a shortcut encourages the diffusion action modeling process to rely on leaked action information instead of learning its own action modeling capability, ultimately leading to inferior performance.
>     ```text
>     <vision> + <language prompt> + <continuous robot state> + <discrete action> + <BOD> + <diffusion noised actions> + <EOD>
>      ```
>
> Exploring the token sequence is a crucial step in our effort to unify diffusion- and autoregressive-based action modeling into a unified model. Using the Type 4 token sequence, we verify that the diffusion and autoregressive learning processes **mutually reinforce each other**, leading to more robust action learning, as demonstrated by the experimental results in Table 3 of the submission. Note that we have also provided a detailed explanation in **our Response Point 1** regarding why the AR action generation is placed after the diffusion generation.
>
> ### **[Updated 2] Modifying Type 3**
> This approach is feasible in practice, but it prevents the `<discrete action>` from being conditioned on the critical `<continuous robot state>` information. Here, we present the results of this experiment, which we refer to as the **“Type 3 - variation.”** As shown in the table below, compared to Type 3, the Type 3 - variant exhibits a noticeable drop in performance for the autoregressive (AR) action outputs, while the diffusion (Dif) actions remain unaffected. This result further suggests that without the temporal consistency provided by the robot state, the AR action generation capability degrades noticeably.
> ||Type 3|Type 3 - variation|
> |-|-|-|
> |AR|0.60|0.54|
> |Dif|0.65|0.66|

---

### Official Review · Reviewer_8izS · 2025-10-31

**Soundness:** 3
**Presentation:** 3
**Contribution:** 3
**Rating:** 6
**Confidence:** 4

**Summary:**

This paper proposes HybridVLA, which is a VLA model that unifies both diffusion of continuous actions and autoregression of discrete action tokens under single LLM architecture. They also propose optimized design choices for the model and collaborative action ensemble for synerging action samples from both processes.

**Strengths:**

- This is one of the first attempts to directly unify diffusion and autoregressive in a VLA model rather than having multiple modules, extending current mainstream VLA design.
- Jointly training diffusion and autoregression improves success rate even without fusion action samples, further improving them with inference time ensemble with confidence threshold.
- The paper reports extensive experiments in simulation and real world robots, and provides meaningful gains over relevant baselines along with multiple experiments and ablations. The results present both practical utility and robustness of the proposed approach.

**Weaknesses:**

- The approach unifies diffusion and autoregression mainly through joint training and an inference-time ensemble, without introducing a deeper fusion or interaction mechanism between the two processes. It raises a minor concern about whether the two processes are truly leveraging each other’s strengths to the maximal extent. We don’t view this as a major weakness, but rather as a potential area for further exploration.

- Naively averaging diffusion and autoregression outputs in collaborative action ensemble could lead to modal conflict when actions are multi-modal. This conflict is well avoided by setting the confidence threshold very high (>0.9) in this work, but there could be more robust methods to fuse action samples even when confidence is low. For example, by using diffusion modal to filter autoregressive actions during inference, which boosts relative confidence of filtered AR action samples, then fusing them.

**Questions:**

- What were specific optimization challenges for jointly training diffusion and autoregression with scalable robotics data mixture and model except token sequence formulation design?
- What are failure cases of setting a low threshold and success cases of setting a higher threshold? It would be persuading to see failures with modal conflict in a 'push-t'-like environment with low threshold.
- In the generalization experiments, using height as the axis for spatial generalization may not convincingly demonstrate a robust generalization for robotic tasks, while positional generalization seems to give us more insight.

---

> ### Author Response · Authors · 2025-11-23
> **Reply to W1**
>
> First of all, thank you for recognizing that our HybridVLA “first attempts to directly unify diffusion and autoregressive methods in a VLA model” and that it provides “both practical utility and robustness.” We also sincerely appreciate your thoughtful comments and questions, which we address in detail below.
>
>
> ## [W1.] Clarification of the Minor Concern
>
> First, we completely agree with your insightful view that exploring how to fully leverage the strengths of both autoregressive and diffusion paradiam for action modeling is an exciting and valuable direction for the robotics community. We will continue to extend our fusion strategy following your suggestions. At the same time, our work represents an early but substantial step toward unifying these two paradigms in single VLA model, and we have made concrete progress toward this goal. We would like to further clarify the importance and effectiveness of our unification strategy:
>
> **a. Unified token-sequence formulation.**  We explore how to integrate autoregressive and diffusion-based action modeling within a **single token sequence**. To achieve this, we carefully design the sequence structure (see Section 3.2 and Table 1). Embedding diffusion modeling inside the LLM allows the model to naturally benefit from the LLM’s internet-scale pretrained knowledge. Meanwhile, by introducing specialized tokens and placing autoregressive tokens after the diffusion tokens, the LLM can interpret the diffusion-modeled latents as part of its implicit conditioning signals. For empirical justification, Type 2 and Type 3 in Table 1 both employ joint training with two losses, just like HybridVLA (Type 4), but adopt different token-sequence designs. Compared with baseline Ex3 in ablation Table 3 (which uses only diffusion loss), Type 2 and Type 3 show no performance improvements despite using the joint training strategy. **This demonstrates that joint training alone is insufficient, the hybrid objectives must be paired with the correct unified token-sequence formulation to enable meaningful mutual reinforcement between the two paradigms.**
>
>
>
> **b. Causal attention mask modification.** Meanwhile, we conduct an additional ablation on our proposed token-sequence formulation by modifying the causal attention mask. Specifically, after the diffusion denoising stage, we mask out all diffusion tokens so they cannot serve as conditioning signals for subsequent AR action generation, effectively removing the continuous-contact prior while keeping the joint training paradigm unchanged. We denote this variant as HybridVLA-mask. As shown in the table below, HybridVLA-mask yields a 5% performance degradation relative to HybridVLA, even though both models use the same token sequence and share identical hybrid objectives. **This result further confirms that, within the HybridVLA action representation, the diffusion and autoregressive learning processes mutually reinforce each other rather than functioning as isolated predictors.**
>
> | **HybridVLA** | **HybridVLA-mask** |
> | --- | --- |
> | 0.78 | 0.73 |
>
> **c. PCA distribution justification.** As further evidence, we provide a principal component analysis (PCA) of the latent distribution in Appendix A.2. The results show that, after our fusion-based training, the model produces diffusion and autoregressive features that form tighter intra-class clusters and larger inter-class separation for both Pick and Place actions. **This indicates that our approach not only improves action generation accuracy but also achieves implicit regularization at the deeper feature level**.
>
>
> **d. During inference,** we further explore the complementary behaviors of autoregressive and diffusion policies and employ a simple yet effective ensembling strategy to enhance action robustness. We hope these findings serve as a meaningful step toward more fully leveraging the strengths of both generation paradigms for action modeling.

---

> ### Author Response · Authors · 2025-11-23
> **Reply to W2 and Q1**
>
> ## [W2] Different collaborative manner
> Thank you for your thoughtful suggestion, which provides valuable guidance for further improving HybridVLA. However, we found that directly using diffusion-predicted actions to filter autoregressive (AR) actions during inference is challenging, because the diffusion branch does not naturally produce an explicit action confidence. Therefore, we designed two similar exploratory experiments across 10 simulation tasks.
> ### **1) Verification-based filtering**
> We develop a learning-based action-ensemble method (**HybridVLA-verifier**). Since our model is capable of both diffusion-based and autoregressive action generation, we append an **action-verifier token** after the AR action tokens. This token is trained to indicate whether the predicted AR action is sufficiently reliable and suitable for fusion with the diffusion action. To support this, we construct a balanced dataset of **positive samples** and **negative samples** (including failure cases and conflict cases), labeled as 1 and 0 respectively. During inference, if the predicted verifier value is 1, we fuse the diffusion and AR actions; otherwise, we fall back to using the diffusion action alone. This design provides a reliable and learnable criterion for action ensembling, while requiring only a single additional token output. As shown in the table below, our learning-based variant achieves outstanding performance compared with HybridVLA-dif. These results demonstrate the extensibility of our model, which can serve not only as an action generator but also as a verifier to enable more robust action outputs.
> ### **2) Distance-metric filtering**
> After obtaining the diffusion-predicted and AR-predicted actions, we compute the Euclidean distance between their end-effector positions in the world coordinate. We retain the AR action only when the distance is below 0.05 m, after which we apply the confidence threshold to determine whether action ensembling should be performed. We refer to this method as HybridVLA-dis. We observe that its performance is similar to HybridVLA. We analyze why action conflicts between the AR and diffusion branches occur extremely rarely in our system: **(1) Shared supervision during training.** The AR and diffusion branches are both trained to fit the _same_ ground-truth actions under the same observations, so they do not learn divergent action modes for identical inputs. **(2) Diffusion latent conditioning for AR.** The AR branch takes the diffusion latent as a conditioning signal, which provides an explicit contact prior and aligns the two paradigms during both training and inference.
>
> Finally, we sincerely appreciate your suggestion. We will incorporate the above exploratory fusion experiments into the main paper to prevent any potential action conflicts between the two paradigms.
> ||**HybridVLA**|**HybridVLA-dif**|**HybridVLA-verifier**|**HybridVLA-dis**|
> |-|-|-|-|-|
> |Mean S.R.|0.78|0.72|0.75|0.76|
> ## [Q1]. optimization challenges analysis
> In HybridVLA training, the input token sequence is undoubtedly one of the most critical factors, as it directly influences model optimization. Simply placing diffusion-related tokens into an LLM does not lead to meaningful learning, so a carefully structured sequence is essential to ensure that both paradigms can be jointly optimized.
>
> In scalable robotic datasets, the diversity of embodiment, task environments, and goal definitions leads to highly heterogeneous data distributions. As the first to scale diffusion-based action modeling to LLMs and jointly optimize it with autoregressive actions, we found that the resulting high-dimensional action space is susceptible to noisy gradients. We also observed that this instability is dependent on the quality of pretraining data: the pretraining corpus contains a portion of noisy or low-quality samples, which caused several training failures, including cases where the loss diverged to NaN. To stably complete pretraining, we repeatedly cleaned the data and adopted gradient checkpointing to monitor optimization and automatically roll back to the most recent stable checkpoint whenever instability was detected.
>
> To further investigate optimization challenges during downstream fine-tuning, we conducted an additional experiment evaluating the dynamic weighting between the AR and diffusion losses across 10 simulation tasks. As shown in the table below, we observe that a ratio slightly above 1:1 yields a relatively stable average success rate of approximately 0.78 to 0.80. When the ratio falls below 1:1, performance becomes noticeably worse. These results demonstrate that once a relatively robust pretrained VLA model is obtained, optimization on high-quality downstream tasks becomes more stable. However, it is still necessary to maintain an appropriate balance between the two losses to ensure effective optimization.
> |L(AR):L(Dif)|Mean S.R.|
> |-|-|
> |10:1|0.79|
> |5:1|0.80|
> |2:1|0.78|
> |1:1|0.78|
> |1:2|0.75|
> |1:5|0.77|
> |1:10|0.75|

---

> ### Author Response · Authors · 2025-11-23
> **Reply to Q2 and Q3**
>
> ## [Q2]. Different threshold analysis
>
> Since we do not have the required assets for the Push-T task, we instead adopt an _Unplug Charger_ task (as shown in Appendix A.1 _Fine-grained Task_), which similarly demands precise positional adjustments and fine-grained control. In this setting, we evaluate HybridVLA under both low and high confidence thresholds when fusing autoregressive and diffusion actions, allowing us to examine how the model behaves under different fusion conditions. As shown in the table below, using a low threshold causes the model to fuse autoregressive actions even when they are not sufficiently reliable. This can introduce action conflicts: the autoregressive prediction slightly perturbs the otherwise smooth diffusion trajectory, resulting in small jitters or minor pose deviations during the unplug motion.
>
> In contrast, a higher confidence threshold ensures that fusion occurs only when the autoregressive output is more reliable. Under this configuration, selective fusion produces more stable trajectories and consistently improves unplug success rates.
>
> | | **HybridVLA(confidence=0.9)** | **HybridVLA(confidence=0.96)** | **HybridVLA-dif** | **HybridVLA-ar** |
> | --- | --- | --- | --- | --- |
> | Mean S.R. | 0.45 | 0.70 | 0.60 | 0.20 |
>
> ## [Q3] Genelization experiment
>
> Thank you for your detailed comments. We agree that positional generalization is a more meaningful measure of model capability. Accordingly, the results reported in Table 5 of the revised main paper are all evaluated under diverse tabletop positions (as noted in Training and Evaluation Details, Lines 464–465 of the revised submission).
>
> Following your suggestion, we conduct a more stringent positional generalization evaluation on the pick-and-place task. Specifically, we re-collected 100 demonstrations to ensure a precise-defined training data distribution. The object locations in the training and test sets are sampled from two non-overlapping tabletop regions. We further evaluate this setting using 20 rollouts. As shown in the table below, under this strict positional generalization scenario, HybridVLA still demonstrates relatively stronger generalization capability.
>
> We have also added Appendix C.3 in the revised submission PDF to illustrate the training data collection regions and the corresponding generalization test regions.
>
> | | **HybridVLA** | **Cogact** |
> | --- | --- | --- |
> | Original | 0.85 | 0.75 |
> | unseen position | 0.75(-11.8%) | 0.50(-33.3%) |

---

> ### Author Response · Authors · 2025-11-28
>
> Dear Reviewer 8izS,
>
> Thank you sincerely for your thoughtful review and for recognizing that HybridVLA “first attempts to directly unify diffusion and autoregressive methods in a VLA model” and provides “both practical utility and robustness.”
>
> Following your raised weaknesses and questions, we have provided clear, point-by-point responses, including: (W1) clarification and verification of how the two generation branches mutually reinforce each other, (W2) exploration of different collaboration mechanisms between the two branches, (Q1) detailed analysis of optimization challenges, (Q2) action-conflict experiments under different AR-confidence thresholds, and (Q3) additional positional-generalization experiments.
>
> If our responses and additional experiments have addressed your concerns, we would sincerely appreciate your consideration in raising the rating. Of course, if any concerns remain, we would be more than happy to discuss them further and provide a prompt response. Thank you again for your thoughtful feedback and valuable reviewing time!

---

### Official Review · Reviewer_r7uJ · 2025-11-03

**Soundness:** 4
**Presentation:** 4
**Contribution:** 4
**Rating:** 6
**Confidence:** 3

**Summary:**

This paper introduces HybridVLA, a method that unifies diffusion-based continuous actions and autoregressive discrete actions within a single LLM backbone. Its core contribution is a hybrid, co-optimized objective that combines diffusion-noise MSE with autoregressive cross-entropy. At inference, it uses few-step DDIM sampling and a confidence-weighted fusion mechanism based on autoregressive token probabilities. The model is pre-trained on cross-embodiment trajectories and then fine-tuned, achieving state-of-the-art performance and strong generalization in real-robot experiments.

**Strengths:**

1.	It unifies diffusion and an LLM in a single backbone and sequence. This integrates the LLM’s reasoning with diffusion’s fine-grained control, while avoiding error accumulation and interface mismatches in multi-module pipelines.
2.	In the unified representation, the diffusion branch provides continuous latents that capture contact and trajectory details. The LLM organizes high-level intent using context and world knowledge. They learn under the same conditional distribution and fuse at inference based on autoregressive confidence.
3.	On RLBench, it achieves 78% average success and surpasses strong baselines. Real-robot tests show stable generalization to unseen objects, backgrounds, heights, and lighting.

**Weaknesses:**

While this work presents a promising hybrid framework, its primary weakness lies in the insufficient justification for its core collaborative mechanism and a lack of sensitivity analysis for its key hyperparameters. The paper claims that hybrid training leads to semantic collaboration, but this could more likely be a regularization effect from the complex loss function, rather than genuine mutual reinforcement between the two paradigms.

**Questions:**

1. How are the weights between diffusion-noise MSE and AR cross-entropy set in the hybrid loss? Please state the basis for weight normalization or dynamic weighting, and report the weight–success-rate curve.
2. How is the confidence threshold for fusion determined and calibrated? Please add a sensitivity analysis for the threshold and the temperature hyperparameter.
3. Table 3 shows that training with the hybrid loss improves performance even when inference uses a single path. How do you establish that this is semantic collaboration?
4. The collaborative fusion relies on a confidence threshold as high as 0.96 to admit AR actions. Please report, per task, the proportion of time steps triggering this threshold, the fusion frequency, and performance under different thresholds. If most steps fall back to pure diffusion, please quantify whether the hybrid mechanism still provides meaningful value.
5. The sequence design places diffusion tokens before AR tokens to prevent leakage, but this also introduces strong causal dependence. When noise is injected into the diffusion branch, does it affect the AR branch and overall decisions?

---

> ### Author Response · Authors · 2025-11-23
> **Reply to W1, Q1, and Q2**
>
> First of all,  thank you for recognizing that our HybridVLA model is “a promising hybrid framework” with unified sequence and representation design.
>
> ## [W1.]
>
> We sincerely appreciate your thoughtful comments and questions. In our responses to Q1 and Q2, we have added sensitivity analyses for the key hyperparameters. In our responses to Q3–Q5, we have provided additional justification for our core collaborative mechanism. These responses correspond to the questions you raised below.
>
> ## [Q1.]  Weights between diffusion and AR losses
>
> Thank you for your constructive comments. Following your suggestion, we conducted a detailed ablation study to examine how the dynamic weighting between the two losses influences manipulation success rates across 10 simulation tasks. Except for adjusting the loss ratios, all other training settings remain identical to those used in the main paper. As shown in the table below, since our model is pretrained on large-scale robotic datasets, the initial values of the two losses are similar. Based on the experimental results in the table below, we provide the following analysis: First, we observe that a ratio between AR and Diff models slightly above 1:1 yields a relatively stable average task success rate of approximately 0.78 to 0.80. When the ratio falls below 1:1, the performance becomes comparatively poorer. Additionally, we find that maintaining a ratio slightly above 1:1 leads to a marginally faster convergence speed during model training. We have also added this ablation study to Appendix C.2 in the revised submission PDF, with the revised parts highlighted in blue.
>
> | L(AR):L(Dif) | Mean S.R. |
> | --- | --- |
> | 10:1 | 0.79 |
> | 5:1 | 0.80 |
> | 2:1 | 0.78 |
> | 1:1 | 0.78 |
> | 1:2 | 0.75 |
> | 1:5 | 0.77 |
> | 1:10 | 0.75 |
>
> ## [Q2.] The confidence threshold and temperature hyperparameter
>
> **Confidence threshold hyperparameter.** Thank you for your detailed comments. In Appendix Table 10, we present an ablation study examining the impact of the confidence threshold on assembly performance (Lines 1151–1162 of the revised submission). We additionally report the per-task scores under each confidence threshold, as shown in the table below. Across different tasks, we find that setting the threshold to **0.96** consistently ensures stable action ensembling, making it a relatively robust threshold setting.
>
> | Threshold | Close box | Close laptop | Toilet seat down | Sweep to dustpan | Close fridge | Phone on base | Take umbrella out of stand | Take frame off hanger | Place wine at rack | Water plants | Mean |
> |:-----------------:|:------------:|:---------------:|:-------------------:|:-------------------:|:---------------:|:----------------:|:------------------------------:|:------------------------:|:---------------------:|:---------------:|:----:|
> | HybridVLA (7B) – 0.90 | 0.80 | 0.85 | 0.95 | **0.95** | 0.85 | 0.50 | 0.40 | 0.55 | 0.55 | 0.45 | 0.68 |
> | HybridVLA (7B) – 0.92 | **0.95** | 0.85 | **1.00** | 0.90 | 0.90 | 0.40 | 0.40 | **0.70** | **0.60** | 0.45 | 0.72 |
> | HybridVLA (7B) – 0.94 | **0.95** | 0.90 | **1.00** | 0.90 | 0.95 | **0.55** | 0.50 | 0.65 | 0.55 | 0.50 | 0.75 |
> | HybridVLA (7B) – 0.96 | **0.95** | **0.95** | **1.00** | 0.90 | **1.00** | **0.55** | **0.60** | **0.70** | **0.60** | **0.55** | **0.78** |
> | HybridVLA (7B) – 0.98 | **0.95** | 0.90 | 0.95 | 0.90 | 0.95 | **0.55** | 0.50 | **0.70** | 0.55 | 0.45 | 0.74 |
>
> **Temperature hyperparameter.** Meanwhile, following your suggestion, we added a sensitivity analysis for the LLM temperature hyperparameter across 10 simulation tasks. It is worth noting that some autoregressive VLA models do not explicitly set the temperature and instead directly select the token with the highest probability (Kim et al., 2024). Our experiments in the main paper also follow this setting. As shown in the table below, we observe that when the temperature is less than or equal to 1, the manipulation success rate remains consistent. However, when the temperature exceeds 1, the action predictions become noticeably unstable. We observe that the robot arm may performs anomalous steps during closed-loop control, ultimately leading to a degradation in accuracy. The results demonstrate that, for robotic tasks, stability in action generation is far more important than output diversity. Therefore, it is reasonable to either adopt the OpenVLA strategy or use a relatively small temperature hyperparameter. We have also added the above meaningful exploration suggestions to Appendix C.2 in the revised submission PDF.
>
> | Temperature | No sample | 0.1 | 0.2 | 0.5 | 1.0 | 1.5 | 2.0 |
> | --- | --- | --- | --- | --- | --- | --- | --- |
> | HybridVLA | 0.78 | 0.78 | 0.77 | 0.78 | 0.76 | 0.71 | 0.64 |

---

> ### Author Response · Authors · 2025-11-23
> **Reply to Q3**
>
> ## [Q3.] Hybrid loss training with single-path inference
>
> We address this question from four perspectives. We appreciate your insightful comments, which helped us further validate our contributions and improve the quality of the paper.
>
> **1) Regularization effect between the two losses.**
> As you pointed out, the two losses introduce a regularization effect when AR and diffusion jointly generate actions. As described in Lines 265–269 of the revised submission, although the two branches differ in their loss functions, both aim to approximate the _same conditional action distribution_ from demonstrations. The action data are normalized identically (range [-1, 1]), and the discrete autoregressive action is simply a quantized representation of this distribution. Therefore, combining the two objectives naturally regularizes the action learning process.
>
>
>
> **2) Mutual reinforcement analysis: changing AR generation from discrete actions to language .**
> To demonstrate that our unified generation paradigm yields mutual reinforcement rather than merely acting as loss regularization, we modify the AR outputs from discrete actions to **language-based sub-goal plans**, as described in Lines 364–369 of the revised submission. The sub-goal plans are manually annotated and further augmented using GPT, and we refer to this variant as **HybridVLA-lang**. For clarity, we summarize the relevant experiments from the submission paper in the table below. We find that HybridVLA-lang achieves a **74%** mean manipulation success rate, outperforming HybridVLA-dif (**72%**). Crucially, the baseline without the hybrid objectives, which relies solely on diffusion for both training and inference, achieves only 65% success (Ex3 in Table 3). These findings show that even when the AR branch predicts sub-goal plans that share only the **_**semantic intent**_** (e.g., “grasp the lid of the box”) but do not model the same action distribution, the two paradigms still reinforce each other and jointly deliver superior performance.
>
> | | **Ex3 in Table 3** | HybridVLA-dif | **HybridVLA-lang** |
> | --- | --- | --- | --- |
> | Mean S.R | 0.65 | 0.72 | 0.74 |
>
>
> **3) Mutual reinforcement analysis: hybrid loss under different token-sequence formulations.**
>
> To further demonstrate that the effectiveness of our unified generation paradigm is not merely attributable to regularization, we evaluate several alternative token-sequence designs (Table 1 of the submission). Both Type 2 and Type 3 incorporate two losses, similar to HybridVLA, yet they perform worse compared to the baseline Ex3 (65% in ablation Table 3), which relies solely on the diffusion loss. This indicates that the hybrid loss must be **paired with the correct token-sequence formulation** in order to yield mutual reinforcement between the two paradigms, thereby improving the stability of manipulation.
>
>
> **4) Mutual reinforcement analysis: hybrid loss with different attention masks.**
> Finally, we conduct an additional experiment on our proposed token sequence by modifying the causal attention mask. Specifically, after diffusion denoising, we mask out the diffusion tokens so that they cannot be used as conditioning signals for AR action generation, thereby removing the continuous contact prior while keeping the hybrid loss unchanged. We refer to this variant as **HybridVLA-mask**. As shown in the table below, HybridVLA-mask exhibits an 5% performance drop compared to HybridVLA, despite both models sharing the same token sequence and hybrid objectives. This provides further evidence that the performance gains arise from the latent conditioning that strengthens action generation, rather than from loss regularization alone.
>
> | Model  | Close box | Close laptop | Toilet seat down | Sweep to dustpan | Close fridge | Phone on base | Take umbrella out of stand | Take frame off hanger | Place wine at rack | Water plants | Mean |
> |:-----------------:|:------------:|:---------------:|:-------------------:|:-------------------:|:---------------:|:----------------:|:------------------------------:|:------------------------:|:---------------------:|:---------------:|:----:|
> | HybridVLA | 0.95 | **0.95** | **1.00** | **0.90** | **1.00** | **0.55** | **0.60** | **0.70** | 0.60 | **0.55** | **0.78** |
> | HybridVLA-mask | **1.00** | 0.90 | 0.90 | 0.80 | 0.95 | 0.45 | 0.55 | 0.65 | **0.65** | 0.45 | 0.73 |

---

> ### Author Response · Authors · 2025-11-23
> **Reply to Q4 and Q5**
>
> ## [Q4]. The detailed confidence threshold and fusion frequency
>
> As shown in our Q2 response, we report the performance of each task under different confidence thresholds. Meanwhile, following your suggestion, we additionally examine the fusion frequency between AR and diffusion actions when the confidence threshold is set to 0.96. As shown in the table below, HybridVLA triggers the collaborative action ensemble strategy in more than 80% of the time steps on average. These results indicate that the chosen threshold of 0.96 is appropriate, as action ensembling is frequently activated throughout execution and effectively enhances the stability of action generation.
>
> |  | Close box | Close laptop | Toilet seat down | Sweep to dustpan | Close fridge | Phone on base | Take umbrella out of stand | Take frame off hanger | Place wine at rack | Water plants | Mean |
> |:-----------------:|:------------:|:---------------:|:-------------------:|:-------------------:|:---------------:|:----------------:|:------------------------------:|:------------------------:|:---------------------:|:---------------:|:----:|
> | Frequency | 0.77 | 0.97 | 0.93 | 0.93 | 0.87 | 0.79 | 0.67 | 0.66 | 0.77 | 0.83 | 0.82 |
>
>
> ## [Q5]. The diffusion noise influence
>
> Thank you for your detailed comments. First, noise in the diffusion branch does not significantly affect AR decisions, because the AR branch has already been exposed to the corresponding diffusion latents during training. Meanwhile, during inference, AR tokens are conditioned only on the denoised diffusion latent, which greatly reduces the influence of noise on AR action generation.
>
> Second, empirically, AR decisions remain stable even when diffusion tokens serve as explicit conditioning signals. As shown in our Q3.4) additional experiment, we modify the causal attention mask by masking out the diffusion tokens after denoising so that they cannot be used as conditioning for AR action generation. We observe that **HybridVLA-mask** leads to an **5%** performance drop, demonstrating that the denoised diffusion tokens provide a beneficial conditioning signal for overall decision-making.

---

> ### Author Response · Authors · 2025-11-28
>
> Dear Reviewer r7uJ,
>
> Thank you sincerely for your insightful review and for recognizing our work as “a promising hybrid framework” with a unified sequence and representation design.
>
> Following the weaknesses and questions you raised, we have provided point-by-point responses, including: (Q1) exploration of weighting strategies between diffusion and AR losses, (Q2) experiments analyzing the impact of confidence-threshold and temperature hyperparameters, (Q3) evaluation of hybrid-loss effects and semantic collaboration, (Q4) a detailed study of AR confidence thresholds and fusion frequency, and (Q5) an explanation of the influence of diffusion noise.
>
> We hope our rebuttal resolves your questions, and would be grateful if you could consider updating the score if our responses are satisfactory. If any questions remain, we would be more than happy to discuss and provide a prompt response. Thank you again for your valuable feedback!

---

### Official Review · Reviewer_bg8d · 2025-11-05

**Soundness:** 3
**Presentation:** 3
**Contribution:** 2
**Rating:** 4
**Confidence:** 3

**Summary:**

The paper proposes HybridVLA, a unified vision-language-action (VLA) model that integrates diffusion-based continuous action generation and autoregressive (AR) discrete action generation within a single LLM backbone. The core idea is a “collaborative training recipe” that inserts diffusion denoising steps into the LLM’s token sequence using special markers (BOD/EOD) so that denoising is treated as iterative “reasoning” steps. A hybrid objective (MSE for diffusion noise prediction + cross-entropy for AR tokens) is optimized jointly, and an inference-time ensemble fuses diffusion and AR actions based on AR token “confidence” with a fixed threshold.
The model is pretrained on ~760k trajectories and then fine-tuned on RLBench and real world tasks. The results shows that the model can achieves better results compared with the pi0 and CogACT baselines.

**Strengths:**

The paper is well-written and easy to follow. The idea of combining AR (semantic reasoning, sample efficiency) with diffusion (continuous, precise control) in one model makes sense to me. The unification via token-sequence design and the BOD/EOD markers is conceptually neat, and the hybrid objective plus confidence-based ensemble is practical. Overall I think this is a promising direction to pursue.

**Weaknesses:**

However, I feel like the current results are not significant enough:
- In the real world experiments, using a 7B model to compare with a 2.7B pi0 baseline seems unfair to me and I did not see an explanation on it (since the authors also have a 2.7B model). It’s also not clear to me to what extent using different pretraining data would affect the final performance.
- The generalization part, which is critical for robotic applications, follows this setting and only pick one scenario (and they are not the same) for each model. Also HybridVLA does not seem more robust compared to the baseline in many cases (similar performance percentage drop).
- In the RLBench experiments, while the full HybridVLA 7B model outperform the HybridVLA-dif variant, it is 50% slower, so it is hard to justify whether using a larger diffusion-only model with the same speed can be on par or even surpass the proposed HybridVLA setting.

**Questions:**

The 0.96 threshold seems adhoc to me, how robust this threshold is (when change backbone / task etc.)?

---

> ### Author Response · Authors · 2025-11-23
> **Reply to W1**
>
> First of all, thank you for recognizing that our idea makes sense to you and represents a promising direction to pursue. We also sincerely appreciate your thoughtful comments and questions, which we address in detail below.
>
> ## [W1.1] Additional real world comparizon with 2.7B model
>
> Thank you for your constructive comments. We employ the 2.7B Phi-2 model as the LLM backbone and compare it with the π₀ model in real-world dual-arm experiments. The dual-arm platform is selected because π₀ demonstrates relatively strong performance in this configuration. The implementation details remain consistent with those described in our main paper. As shown in Table below, the results follow a similar trend to those in the simulation experiments (Table 2 in the main paper). HybridVLA (2.7B) achieves consistently superior results compared to π₀, validating the effectiveness of our proposed method and demonstrating its generalizability across different backbones. Finally, we would like to clarify that we selected LLaMA-2 7B as the backbone for our main experiments because it allows a fair comparison with other baseline VLA models such as (Li et al., 2024b), (Kim et al., 2024), and (Li et al., 2024a), which are all built on the same LLM. In contrast, π₀ is based on the PaliGemma LLM, whose performance is generally stronger than LLaMA-2 on many standard benchmarks.
>
> ## [W1.2] The impact of using different pretraining data.
>
> This is an interesting question. Based on our extensive empirical experience with VLA pretraining, using different pretraining datasets indeed influences downstream task performance. Two factors are particularly critical: (1) the similarity between the robot embodiments and tasks in the pretraining data and those in the downstream tasks, and (2) the quality and quantity of the data. Similar to other VLA papers (Liu et al., 2024b) (Kim et al., 2024), the exploration and cleaning of pretraining datasets represent one of our contributions.
>
> To conduct a more comprehensive comparison with π₀, we note that the original training data of π₀ are not publicly available, as they rely heavily on privately collected datasets. Therefore, we re-pretrained π₀ on our own pretraining dataset for the same number of epochs as HybridVLA to obtain **π₀ (pre)**. As shown in the last row of the below table, π₀ (pre) achieves comparable performance to the original π₀. These findings indicate that while pretraining data critically shape a model’s manipulation capability, when pretrained data quality are adequate and all models are fine-tuned on a high-quality downstream dataset, the resulting performance differences are not significant.
>
> | | Pick and place | Lift ball and place it in basket | Place two bottles at rack | Wipe blackboard using erase | Fold shorts | Mean S.R. |
> | --- | --- | --- | --- | --- | --- | --- |
> | π₀ | 0.75 | 0.65 | 0.40 | 0.30 | 0.65 | 0.55 |
> | HybriVLA(2.7B) | 0.80 | 0.70 | 0.65 | 0.40 | 0.65 | 0.64 |
> | HybriVLA(7B) | 0.90 | 0.80 | 0.60 | 0.55 | 0.70 | 0.71 |
> | π₀(pre) | 0.60 | 0.65 | 0.45 | 0.35 | 0.60 | 0.53 |

---

> ### Author Response · Authors · 2025-11-23
> **Reply to W2**
>
> ## [W2] Additional generalization experiments
>
> Thank you for your insightful comments. We would like to clarify the rationale behind our selection of generalization scenarios. For the single-arm evaluation, we selected the classic _Pick and Place_ task, which allows for constructing a wide range of unseen variations. For the dual-arm evaluation, we selected the _Lift Ball and Place_ task, as it highlights the coordination required in dual-arm manipulation. While the accuracy drop percentage is occasionally comparable to that of **π₀** in the main paper for some configurations, we completely agree with your insightful suggestion that assessing generalization ability based on a single task alone is not sufficiently comprehensive.
>
> **Following your suggestion, to enable a more comprehensive generalization comparison, we introduced two new dual-arm generalization tasks: _Pick and Place_, and _Place Two Bottles on Rack_, as shown in the table below**. Specifically, for the dual-arm _Pick and Place_ task, in the **unseen object** scenario, we replaced the manipulated objects with other fruits (e.g., oranges and cucumbers); for _Place Two Bottles on Rack_, we replaced the manipulated objects with bottles of different shapes and colors, while keeping other unseen configurations consistent with those in generalization experiments of main paper.
>
> To provide a clearer overview, we summarize in the table below the **average score** and **average accuracy drop percentage** across all unseen configurations. The results show that our method reduces the accuracy drop by approximately **5–16%** compared to the baselines under generalization scenarios.
>
> These findings demonstrate that **HybridVLA effectively integrates diffusion into the autoregressive next-token prediction process**, achieving not only **more robust action generation**, but also **more efficient learning from demonstrations**, thereby enhancing its generalization capability across diverse tasks.
>
> We have accordingly updated Section 4.4 and Table 6 in the submission PDF, with the revised parts highlighted in blue, and we will further include additional generalization tasks in the revised version.
>
>
> | Dual arm: Lift ball and place | Original  | Object | Background | Height | Lightning | Mean |
> | --- | --- | --- | --- | --- | --- | --- |
> | HybridVLA | 0.80 | 0.75(-6%) | 0.60(-25%) | 0.60(-25%) | 0.75(-6%) | 0.68(-15%) |
> | π₀ | 0.65 | 0.60(-8%) | 0.50(-23%) | 0.45(-31%) | 0.55(-15%) | 0.52(-20%) |
>
>
> | Dual arm: Pick and Place  | Original  | Object | Background | Height | Lightning | Mean |
> | --- | --- | --- | --- | --- | --- | --- |
> | HybridVLA | 0.90 | 0.90(-0%) | 0.80(-11%) | 0.70(-22%) | 0.80(-11%) | 0.80(-11%) |
> | π₀ | 0.75 | 0.55(-26%) | 0.50(-33%) | 0.50(-33%) | 0.65(-13%) | 0.55(-27%) |
>
>
> | Dual arm: Place Two Bottles on Rack | Original  | Object | Background | Height | Lightning | Mean |
> | --- | --- | --- | --- | --- | --- | --- |
> | HybridVLA | 0.60 | 0.55(-8%) | 0.50(-17%) | 0.45(-25%) | 0.55(-8%) | 0.51(-15%) |
> | π₀ | 0.40 | 0.30(-25%) | 0.30(-25%) | 0.25(-37%) | 0.35(12.5%) | 0.30(-25%) |

---

> ### Author Response · Authors · 2025-11-23
> **Reply to W3 and Q1**
>
> ## [W3] Additional Comparison Between HybridVLA and HybridVLA-dif
>
> We sincerely appreciate your valuable comments, which help us conduct a more comprehensive validation of our method’s effectiveness. Since robotic manipulation requires relatively efficient inference, we follow **GR00T N1 (Björck et al., 2025)** by utilizing **middle-layer instead of final-layer LLM embeddings**. Specifically, we directly extract middle-layer features from **LLaMA-2 7B** for action generation (HybridVLA-half), achieving a similar inference speed to **HybridVLA-dif (7B)**. As shown in the table below, HybridVLA-half still delivers strong performance. These results demonstrate that robust action generation does not depend on increased computational cost, but rather on our method to effectively integrate two generation paradigms within a shared LLM backbone.
>
> | | HybridVLA-dif (7B) | HybridVLA-half (7B) | HybridVLA (7B) |
> | --- | --- | --- | --- |
> | Mean S.R. | 0.72 | 0.76 | 0.78 |
>
>
> Moreover, the **HybridVLA-dif** variant is also a contribution of our paper, as it is jointly trained using our proposed collaborative training strategy but relies solely on diffusion-based action generation during inference (Line 347 - 349 of the revised submission). Through this variant, we aim to show that our method can preserve the continuous characteristics of diffusion based actions while also inheriting the pretrained autoregressive generation paradigm of the LLM, allowing the model to learn efficiently from demonstrations during training, even when only diffusion based action outputs are used at test time. Compared with existing diffusion-based VLA approaches that simply attach a diffusion head after the VLM, our method more effectively leverages the pretrained knowledge within the LLM to better unlock the potential of diffusion-based action prediction.
>
>
> ## [Q1] Additional exploration with AR confidence threshold
>
> Thank you for your detailed question. In Appendix Table 10, we provide an ablation study examining the impact of the confidence threshold on assembly output performance (Line 1152-1162 of the revised submission). **For different tasks,** we additionally report the score of each task under every confidence threshold, as shown in the table below.
>
> **For different backbones**, following your suggestion, we conducted the same ablation experiment using the 2.7B Phi-2 model as the LLM backbone. As shown in the table below, we find that setting the action-token confidence threshold to 0.96 still serves as a robust indicator for determining whether action ensembling should be applied, regardless of the underlying LLM backbone. We have also added this ablation study to Section 4.2 in the revised submission PDF.
>
> | Model - Threshold | Close box | Close laptop | Toilet seat down | Sweep to dustpan | Close fridge | Phone on base | Take umbrella out of stand | Take frame off hanger | Place wine at rack | Water plants | Mean |
> |:-----------------:|:------------:|:---------------:|:-------------------:|:-------------------:|:---------------:|:----------------:|:------------------------------:|:------------------------:|:---------------------:|:---------------:|:----:|
> | HybridVLA (7B) – 0.90 | 0.80 | 0.85 | 0.95 | **0.95** | 0.85 | 0.50 | 0.40 | 0.55 | 0.55 | 0.45 | 0.68 |
> | HybridVLA (7B) – 0.92 | **0.95** | 0.85 | **1.00** | 0.90 | 0.90 | 0.40 | 0.40 | **0.70** | **0.60** | 0.45 | 0.72 |
> | HybridVLA (7B) – 0.94 | **0.95** | 0.90 | **1.00** | 0.90 | 0.95 | **0.55** | 0.50 | 0.65 | 0.55 | 0.50 | 0.75 |
> | HybridVLA (7B) – 0.96 | **0.95** | **0.95** | **1.00** | 0.90 | **1.00** | **0.55** | **0.60** | **0.70** | **0.60** | **0.55** | **0.78** |
> | HybridVLA (7B) – 0.98 | **0.95** | 0.90 | 0.95 | 0.90 | 0.95 | **0.55** | 0.50 | **0.70** | 0.55 | 0.45 | 0.74 |
> |  |  |  |  |  |  |  |  |  |  |  |  |
> | HybridVLA (2.7B) – 0.90 | 0.70 | 0.75 | 0.85 | **0.80** | **0.90** | **0.25** | 0.45 | 0.40 | 0.50 | 0.10 | 0.58 |
> | HybridVLA (2.7B) – 0.92 | 0.85 | **0.90** | 0.90 | **0.80** | 0.85 | **0.25** | 0.45 | 0.35 | 0.50 | 0.20 | 0.61 |
> | HybridVLA (2.7B) – 0.94 | **1.00** | 0.85 | 0.95 | 0.75 | 0.85 | **0.25** | 0.40 | 0.40 | 0.60 | 0.25 | 0.63 |
> | HybridVLA (2.7B) – 0.96 | **1.00** | **0.90** | 0.90 | **0.80** | **0.90** | **0.25** | **0.55** | **0.45** | **0.70** | **0.25** | **0.67** |
> | HybridVLA (2.7B) – 0.98 | 0.90 | **0.90** | **0.95** | 0.55 | **0.90** | 0.20 | **0.55** | 0.35 | **0.70** | 0.15 | 0.62 |

---

> ### Author Response · Authors · 2025-11-28
>
> Dear Reviewer bg8d,
>
> Thank you sincerely for your thoughtful feedback. We truly appreciate your positive recognition that “our idea makes sense and represents a promising direction to pursue.”
>
> We have provided comprehensive responses to all raised concerns, including: (W1) additional real-world comparisons using the 2.7B model and experiments with different pretraining data, (W2) expanded generalization evaluations, (W3) further comparison between HybridVLA and HybridVLA-dif, and (Q1) a detailed exploration of AR confidence–threshold behaviors across tasks and backbones.
>
> We hope our responses can address your concerns, and we would be grateful if you could consider updating the score if you find our response satisfactory. We remain available for any further discussion should there be unresolved concerns. Thank you again for your valuable review time!

---

### Author Response · Authors · 2025-11-30
**Overall Rebuttal Summary**

Dear Reviewers, Area Chairs, and Program Chairs,

First and foremost, we would like to express our sincere appreciation for your valuable time and dedicated effort. We are encouraged that HybridVLA received relatively positive recognition, and we are grateful that three key aspects of our work were explicitly acknowledged in the reviews:
- **Method and novelty:** `Reviewer bg8d` noted that "the idea of combining AR with diffusion in one model **makes sense** to me". `Reviewer r7uJ` described HybridVLA as "a **promising hybrid framework** that integrates the LLM’s reasoning with diffusion’s fine-grained control". `Reviewer 8izS` recognized that HybridVLA "**first attempts** to directly unify diffusion and autoregressive methods in a VLA model" and that "jointly training diffusion and autoregression **improves success rate**". `Reviewer 4X5g` highlighted that "this paper aims to design a **novel policy architecture**".
- **Evaluation:** "achieves better results" (`Reviewer bg8d`); "surpasses strong baselines and shows stable generalization" (`Reviewer r7uJ`); "provides meaningful gains, and the results present both practical utility and robustness" (`Reviewer 8izS`); "shows promising results against other VLA variants" (`Reviewer 4X5g`).
- **The importance to the community:** `Reviewer bg8d` described our approach as "a promising direction to pursue." `Reviewer 8izS` noted that HybridVLA "extends current mainstream VLA design," and `Reviewer 4X5g` emphasized that the paper addresses "a valuable problem" and explores "a promising direction."

Meanwhile, we carefully reviewed all comments and provided detailed point-by-point responses. As the discussion period ended unexpectedly, we summarize below the key points and additional analyses from our rebuttal to help Reviewers and Area Chairs clearly understand our updates.
- `Reviewer bg8d`’s concerns focused on experimental evaluation, which we addressed with additional experiments and clarifications. We added real-world comparisons using the 2.7B HybridVLA, analyzed the effect of different pretraining datasets (W1), expanded generalization evaluations (W2), conducted further comparisons between HybridVLA and HybridVLA-dif (W3), and provided a detailed analysis of AR confidence–threshold behavior across tasks and model backbones (Q1).
- `Reviewer r7uJ` mainly requested additional ablations with deeper analysis. In response, we performed sensitivity studies on key hyperparameters and further justified our collaborative mechanism, including: (Q1) weighting strategies between diffusion and AR losses, (Q2) effects of confidence thresholds and temperature, (Q3) hybrid-loss and semantic collaboration analysis, (Q4) AR confidence–threshold and fusion-frequency behavior, and (Q5) the influence of diffusion noise.
- `Reviewer 8izS` requested deeper analysis of HybridVLA and additional exploratory experiments. In response, we provided clear clarifications, including: (W1) verification and explanation of how the two generation branches mutually reinforce each other, (W2) exploration of alternative collaboration mechanisms, (Q1) analysis of optimization challenges, (Q2) action-conflict experiments under different AR-confidence thresholds, and (Q3) additional positional-generalization evaluations.
- `Reviewer 4X5g` initially found the rationale and technical contributions of unifying autoregressive and diffusion action generation within a unified HybridVLA framework unclear. To address this, we provided additional explanations and supplementary experiments demonstrating the soundness and originality of HybridVLA’s design. As a result, **`Reviewer 4X5g` expressed "I will increase the score to borderline" and raised further questions**, primarily concerning the specific implementation details—namely, the roles and ordering of the autoregressive and diffusion outputs. In response, we clarified the full process step-by-step, from model inputs to design rationale, and supplemented our explanation with additional experimental results and analysis. It is unfortunate that `Reviewer 4X5g` was unable to continue the discussion, as we are confident that further clarification would have allowed them to fully appreciate the technical contributions of HybridVLA.

We hope that the clarifications, additional analyses, and newly introduced experiments have adequately addressed the Reviewers’ main concerns. We have incorporated all major points discussed during the rebuttal into the main paper and appendix, with all revisions highlighted in blue for ease of reference. **We respectfully ask that the considerations raised during the rebuttal—together with the improvements made in direct response to each comment—be fully taken into account when assessing HybridVLA’s merit and contribution.** We again thank the Reviewers for their time and constructive feedback, and we are grateful to the Area Chairs and Program Chairs for their continued evaluation.

Best regards,

Authors of Submission 12341

---

### Meta-Review · Area_Chair_wrBH · 2026-01-07

**Summary:**

The paper proposes a hybrid VLA model, in which the VLM predicts both continuous diffusion action tokens and discrete AR action tokens instead of a separate diffusion head as done in conventional methods. The paper uses a collaborative training method and collaborative ensemble in which diffusion and AR outputs are aggregated.

The main concern from reviewers were missing motivation of the hybrid architecture and missing generalization experiments. The authors have sufficiently added a lot of experiments in the rebuttal.
I still believe the paper misses a good motivation of why we need both discrete and continuous action prediction. This objective is not standard, hence it merits proper explanation and analysis.

While it is true that we miss clear motivation, empirical experimental results are quite good. The authors did a great job in the rebuttal conducting a lot of experiments. Due to solid experiments and good empirical results, I vote for accepting the paper.

**Reviewer Concerns:**

The authors addressed most of the experiment concerns in the rebuttal. Motivation is not sufficiently addressed.

**Reviewer Scores:**

Reviewer bg8d raises concerns about comparison with smaller \pi model being unfair, and poor generalization experiments. Authors included experiments with 2.5B model, and more generalization experiments. I believe the reviewer will increase their score to 5 or 6 after this.
Reviewers r7uJ is concerned about insufficient justification for its core collaborative mechanism and a lack of sensitivity analysis for its key hyperparameters. The sensitivity experiments are included and some justification is provided. The reviewer might retain their score, which is already positive.
Reviewer 8izS  raised minor concern if naive avergaeing is optimal, and also raised some questions on experimental details, which the authors have addressed.
Finally, reviewer 4X5g raised point mainly about the motivation for hydrid objective. The reviewer misunderstood the training - the authors do not have text prediction following diffusion action outputs, but rather have action tokens itself. For other concerns on novelty clarification, authors included sufficient analysis. So, I think the reviewer would have increased the score to borderline or higher.

---

### Decision · Program_Chairs · 2026-01-26

Accept (Poster)